# DVSOD: RGB-D Video Salient Object Detection

**Jingjing Li**[1], **Wei Ji**[1,2*], **Size Wang**[1], **Wenbo Li**[2], **Li Cheng**[1]

[1]University of Alberta, Canada
[2]Samsung Research America AI Center, USA
{jingjin1, wji3, size1, lcheng5}@ualberta.ca, wenbo.li1@samsung.com

## Abstract

Salient object detection (SOD) aims to identify standout elements in a scene, with recent advancements primarily focused on integrating depth data (RGB-D) or temporal data from videos to enhance SOD in complex scenes. However, the unison of two types of crucial information remains largely underexplored due to data constraints. To bridge this gap, we in this work introduce the DViSal dataset, fueling further research in the emerging field of RGB-D video salient object detection (**DVSOD**). Our dataset features 237 diverse RGB-D videos alongside comprehensive annotations, including object and instance-level markings, as well as bounding boxes and scribbles. These resources enable a broad scope for potential research directions. We also conduct benchmarking experiments using various SOD models, affirming the efficacy of multimodal video input for salient object detection. Lastly, we highlight some intriguing findings and promising future research avenues. To foster growth in this field, *our dataset and benchmark results are publicly accessible at: `https: // dvsod. github. io/`.*

## 1 Introduction

Salient object detection (SOD) is a fascinating field of study that aims at identifying and distinguishing the most eye-catching components in static images [59, 91] or dynamic videos [66, 89]. This intriguing concept hails from cognitive science research into human visual attention behavior - our incredible ability to swiftly direct attention to the most valuable portions of visual scenes [4]. As such, SOD assumes an indispensable role in a myriad of real-world applications [52, 53, 30, 50, 69, 96, 24, 93, 81], such as image retrieval, video editing/analysis, medical diagnosis, object detection, and target tracking.

This field has witnessed significant advances in recent years, particularly in RGB-image based SOD [65, 90, 82, 72]. However, it has been observed that performance often degrades when objects and their surroundings share similar appearances or when backgrounds are heavily cluttered. With the growing ubiquity of 3D imaging sensors in depth cameras, such as Kinect and Intel RealSense, as well as in mobile devices like iPhone 13 Pro, Huawei Mate 50, and Samsung Galaxy S21, the value of fully leveraging RGB-D information for SOD has gained considerable research interest [14, 58, 94, 95]. As visualized in Fig. 1, paired depth maps contribute valuable 3D geometric information that helps the discernment of object silhouettes against backgrounds. Consequently, augmenting RGB images with depth maps as additional input dramatically enhances the accuracy in localizing salient objects in complex scenes, *i.e.*, '*w*/ RGB' *vs.* '*w*/ RGBD' in Fig. 1.

On another front, considering the dynamic nature of real-world scenes, video salient object detection (VSOD) has been a prominent focus of research [19, 56, 71]. In contrast to static images, dynamic video scenario presents considerable difficulties due to the diversity of motion patterns, occlusions, blurring, large object deformations, and the inherent complexity of human visual attention behavior.

---

* Corresponding author.

37th Conference on Neural Information Processing Systems (NeurIPS 2023) Track on Datasets and Benchmarks.

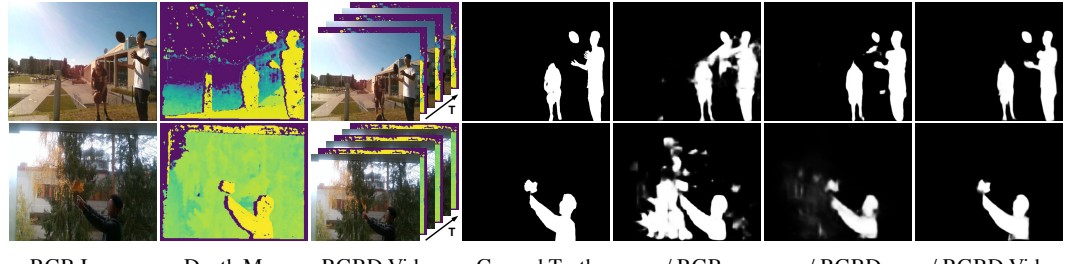

| RGB Image | Depth Map | RGBD Video | Ground Truth | *w/* RGB | *w/* RGBD | *w/* RGBD Video |

Figure 1: Visual illustration of the advantages of employing RGBD videos for salient object detection. The last three columns exhibit the segmentation results achieved using different input modalities.

By leveraging temporal contexts within a video sequence, VSOD methods [19, 63] have demonstrated improved capture of moving foreground objects and modeling of visual attention shifts in dynamic environments.

While there have been significant strides in both RGB-D SOD and VSOD individually, there remains a paucity of research into combining multimodal and temporal contexts, both of which are vital for accurately detecting the salient objects in a scene. This lack is primarily due to the absence of suitable benchmark datasets. As an alternative, Lu *et al.* [48] have made the initial attempt to amalgamate existing VSOD datasets with estimated depth maps. However, these estimated depth maps often fall short of accurately reflecting the real world, and are prone to errors introduced by depth estimation methods, compared to data captured by actual depth sensors.

To bridge the existing gap and prosper the progress in RGB-D video SOD (*i.e.*, DVSOD), we introduce the first real-world RGB-**D Vi**deo **Sal**iency detection dataset, abbreviated as DViSal, in this study. Specifically, the DViSal dataset comprises 237 RGB-D videos at a frame rate of 25 f/s, including 175,442 RGB-D pairs in total and 7,117 annotated frames. Besides, the videos were carefully selected to encompass a broad diversity of real scenarios and motion patterns. Another distinguishing characteristic of the DViSal dataset is the variety of annotation formats provided. In addition to the conventional object-level annotations used in SOD tasks, the dataset offers more comprehensive instance-level annotations, as well as weak annotations consisting of bounding boxes and scribbles. These diverse annotation sets may serve as valuable resources for a wider range of potential research directions, such as instance-level DVSOD, weakly-supervised DVSOD, and more.

We further benchmark the DVSOD by implementing 11 cutting-edge SOD models on the newly-proposed DViSal dataset, including *RGB-image-based* SOD models, *RGB-D* SOD models, and *VSOD* models. Additionally, we have assembled a straightforward DVSOD model to dissect the specific contributions offered by depth maps and video contexts. The empirical results underscore the utility of both the multimodal RGB-D cues and temporal video contexts in accurately identifying salient objects. In particular, the DVSOD model is shown to significantly enhance the performance of the base RGB network [73], as evidenced by a noteworthy increase in the F-measure (*i.e.*, 0.548 to 0.610). Lastly, we contemplate and highlight promising avenues for future research. We observe that the field of DVSOD is far from fully explored, with substantial room for improvement and innovation. This research presents the community with an exciting opportunity to delve deeper into this burgeoning field.

## 2   Related Works

In the era of deep learning, benchmark datasets have become the foundational infrastructure for advancing the state-of-the-art in computer vision research. Prominent publicly available benchmarks, such as DUTS [70], SOC [17], and PASCAL-S [42], have greatly contributed to the remarkable progress in RGB-image-based salient object detection (SOD) [73, 65, 72] over the past decade. Despite their significant contributions, these benchmarks primarily center their attention on RGB images, overlooking the valuable depth data that can enhance the discrimination of object silhouettes from backgrounds.

With the rising popularity of 3D imaging sensors, acquiring RGB-D data has become easier with modern equipments (*e.g.*, Kinect, Samsung Galaxy S21 and iPhone 13 Pro). This development has

Table 1: Overview of existing RGB SOD, RGB-D SOD, VSOD datasets, and our proposed DViSal dataset. The table shows whether each dataset provides depth maps/video sequences (Depth/Video), object-level ground truths (Obj-GT), instance-level annotations (Ins-GT), and weak supervision signals (Weak-GT) such as bounding boxes or scribbles. The table also lists the number of videos (#Video), annotations (#GT), and instances (#Ins) each dataset includes.

| Dataset | Year | Depth | Video | Obj-GT | Ins-GT | Weak-GT | #Video | #GT | #Ins |
|---|---|---|---|---|---|---|---|---|---|
| MSRA-B [46] | 2007 | ✗ | ✗ | ✓ | ✗ | ✓ | - | 5,000 | - |
| SED1&2 [2] | 2007 | ✗ | ✗ | ✓ | ✗ | ✗ | - | 200 | - |
| ASD [1] | 2009 | ✗ | ✗ | ✓ | ✗ | ✗ | - | 1,000 | - |
| SOD [51] | 2010 | ✗ | ✗ | ✓ | ✗ | ✗ | - | 300 | - |
| M10K [8] | 2011 | ✗ | ✗ | ✓ | ✗ | ✗ | - | 10,000 | - |
| DUT-O [76] | 2013 | ✗ | ✗ | ✓ | ✗ | ✓ | - | 5,168 | - |
| ECSSD [74] | 2013 | ✗ | ✗ | ✓ | ✗ | ✗ | - | 1,000 | - |
| PASCAL-S [42] | 2014 | ✗ | ✗ | ✓ | ✗ | ✗ | - | 850 | - |
| HKU-IS [36] | 2015 | ✗ | ✗ | ✓ | ✗ | ✗ | - | 4,447 | - |
| SOS [78] | 2015 | ✗ | ✗ | ✗ | ✗ | ✓ | - | 6,900 | - |
| MSO [78] | 2015 | ✗ | ✗ | ✗ | ✗ | ✓ | - | 1,224 | - |
| DUTS [70] | 2017 | ✗ | ✗ | ✓ | ✗ | ✓ | - | 15,572 | - |
| SOC [17] | 2022 | ✗ | ✗ | ✓ | ✓ | ✓ | - | 6,000 | 5,776 |
| STERE [55] | 2012 | ✓ | ✗ | ✓ | ✗ | ✗ | - | 1,000 | - |
| NJU2K [31] | 2014 | ✓ | ✗ | ✓ | ✗ | ✗ | - | 1,985 | - |
| NLPR [60] | 2014 | ✓ | ✗ | ✓ | ✗ | ✗ | - | 1,000 | - |
| DES [9] | 2014 | ✓ | ✗ | ✓ | ✗ | ✗ | - | 135 | - |
| LFSD [41] | 2014 | ✓ | ✗ | ✓ | ✗ | ✗ | - | 100 | - |
| SSD [97] | 2017 | ✓ | ✗ | ✓ | ✗ | ✗ | - | 80 | - |
| DUT-D [62] | 2019 | ✓ | ✗ | ✓ | ✗ | ✗ | - | 1,200 | - |
| SIP [14] | 2020 | ✓ | ✗ | ✓ | ✓ | ✗ | - | 929 | 1,471 |
| ReDWeb-S [44] | 2021 | ✓ | ✗ | ✓ | ✗ | ✗ | - | 3,179 | - |
| COME [79] | 2021 | ✓ | ✗ | ✓ | ✓ | ✓ | - | 15,625 | 26,862 |
| SegV2 [35] | 2013 | ✗ | ✓ | ✓ | ✗ | ✗ | 14 | 1,065 | - |
| FBMS [56] | 2014 | ✗ | ✓ | ✓ | ✗ | ✗ | 59 | 720 | - |
| MCL [32] | 2015 | ✗ | ✓ | ✓ | ✗ | ✗ | 9 | 463 | - |
| ViSal [71] | 2015 | ✗ | ✓ | ✓ | ✗ | ✗ | 17 | 193 | - |
| DAVIS [61] | 2016 | ✗ | ✓ | ✓ | ✗ | ✗ | 50 | 3,455 | - |
| UVSD [47] | 2017 | ✗ | ✓ | ✓ | ✗ | ✗ | 18 | 3,262 | - |
| VOS [37] | 2018 | ✗ | ✓ | ✓ | ✗ | ✗ | 200 | 7,467 | - |
| DAVSOD [15] | 2019 | ✗ | ✓ | ✓ | ✓ | ✗ | 226 | 23,938 | 39,498 |
| **Our DViSal** | - | ✓ | ✓ | ✓ | ✓ | ✓ | 237 | 7,117 | 20,226 |

spurred the creation of several RGB-D datasets aimed at addressing the segmentation of salient objects in complex scenarios. For instance, the STERE dataset [55], which serves as the first collection of stereoscopic photos in this domain, boasts of 797 samples. The NJU2K dataset [31], on the other hand, houses 1,985 paired images in its most recent version, sourced from various platforms such as the internet, 3D movies, and images captured by a Fuji W3 stereo camera. Large-scale benchmarks such as ReDWeb-S [44] and COME [79] have been curated to validate the scalability of diverse models, thereby fostering advancement in this domain. Many novel RGB-D models [28, 34, 23, 16, 85, 86, 45, 38, 39] have been developed, showing that depth data, which includes spatial structure and 3D layout cues within a scene, can assist segmentation in challenging scenarios. For example, [34] proposed a prototype sampling network to adaptively weight RGB and depth superpixels according to a reliance selection module. [45] explored visual transformer to capture global contexts, achieving promising results.

It is worth noting that the existing RGB-D SOD datasets rely on single static images. The lack of a mechanism to account for the temporal contexts might impede their effectiveness when dealing with video inputs that depict dynamic scenes, which are omnipresent in our daily lives [87, 83, 40, 26]. As depicted in Table 1, the current SOD datasets provide as input either single pairs of RGB and depth images (*e.g.*, STERE [55] and NJU2K [31]), or RGB only video sequences (*e.g.*, VOS [37] and DAVSOD [15]). There unfortunately lacks a suitable dataset to provide both types of critical information - spatial layout cues derived from depth data and temporal context derived from dynamic video sequences, in addition to the RGB images.

In this work, we focus on a systematic exploration of RGB-D video salient object detection to bridge this existing gap. We introduce a carefully curated benchmark dataset named DViSal. Table 1

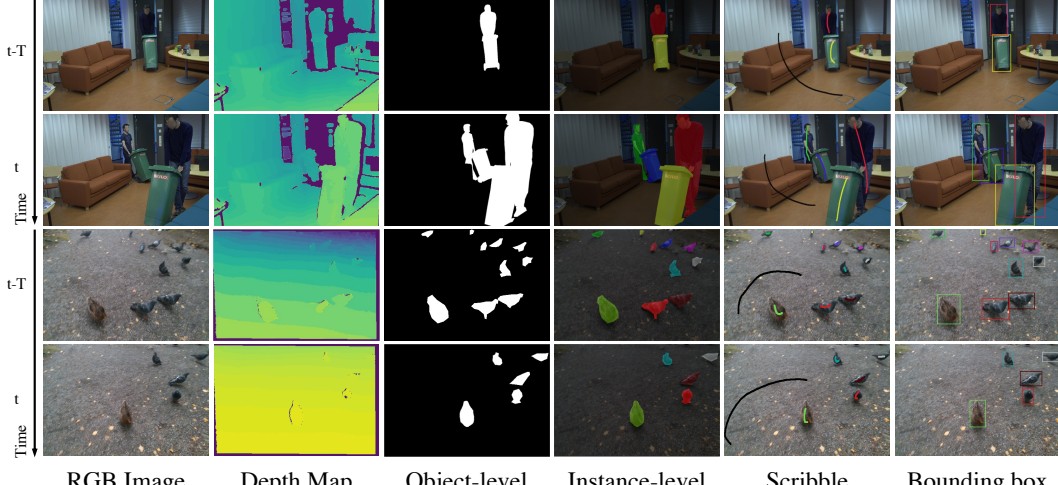

Figure 2: Examples of the DViSal dataset. We provide diverse annotations, including fully-supervised object-/instance-level markings, as well as weakly-supervised scribbles and bounding boxes.

presents the statistics of our newly launched DViSal alongside its counterparts. In addition, we undertake extensive benchmarking experiments that verify the advantages of utilizing RGB-D videos for saliency detection.

## 3   The DVSOD Dataset

In this section, we introduce the new RGB-**D Vi**deo **Sal**iency detection dataset, *i.e.*, ***DViSal***. We first describe the construction process of the DViSal dataset, and then analyze its statistical results.

### 3.1   Dataset Construction

**Dataset Collection.** The main principle of data acquisition is to provide a comprehensive collection of calibrated RGB and depth video sequences, covering a broad spectrum of scenarios. Toward this objective, we initially gathered over 703 RGB-D videos, sourced from multiple repositories, including CDTB [49], People [68], PTB [67], Scene [33], DET [75], Tracklam [5], and Track3D [77]. The collected video samples exhibit diverse locations (*e.g.*, parks, campuses, streets and indoor scenes), and are recorded under a wide range of challenging conditions (*e.g.*, cluttered backgrounds, low-light conditions, and reflective environments). We are committed to ensuring the quality of our dataset. To this end, we meticulously remove unqualified videos or frames - ones that are blurry, misaligned, or feature ambiguous salient objects. After this selection process, we consolidate the DViSal dataset, which consists of 237 high-quality RGB-D videos, with 175,442 paired frames in total. Some visual examples from our dataset are illustrated in Fig. 2.

**Dataset Annotation.** Diverging from most established SOD datasets that primarily offer object-level saliency masks, our proposed dataset aims to deliver more comprehensive annotations. We employ the Labelme toolkit to annotate the collated RGB-D videos. The acquisition of the ground-truth saliency adheres to the annotation principle utilized in widely-recognized SOD datasets [62, 15, 41]. Initially, five annotators select candidate salient objects based on their initial instinct. Subsequently, a majority voting strategy is deployed to finalize the salient objects. Nonetheless, annotating a large-scale DViSal dataset presents additional challenges compared to labeling an RGB image dataset. First, our DViSal dataset includes many challenging scenes recorded under diverse conditions, *e.g.*, multiple objects, transparent objects, similar foreground and background, and cluttered environment. These complexities make it more difficult to identify whole objects and distinguish their silhouettes. Second, for video sequences, it becomes necessary to account for the attention shift [15] of salient objects within a single video. For example, an object labeled as "background" in one frame may be categorized as "salient" in another if the camera view changes. To overcome these challenges, we put into action several proactive measures. We carry out a visualization assistance process that overlays depth heatmaps onto corresponding RGB images, facilitating easier verification of salient

objects in complex scenarios by the annotators. Moreover, for videos showcasing saliency shifting, two additional inspectors are required to review the initial annotations on an item-by-item basis, identifying any mislabeled samples and sending them back to the annotators for corrections and re-verification. Through these efforts, we successfully obtain a large-scale DViSal dataset comprising 7,117 high-quality annotations. These include object-level saliency ground truths, instance-level IDs, and weak supervision signals, such as bounding boxes and scribbles.

**Dataset Splits.** The entire dataset is partitioned into training, validation and test sets, which consist of 103, 26, and 108 videos, respectively, with 3,560, 200, and 3,357 annotated frames each. To be specific, the training set includes the entire CDTB [49] dataset, which contains 71 video sequences, and an additional 32 randomly selected videos from the PTB [67], Tracklam [5] and Track3D [77] datasets. The validation set is made up of 26 videos sourced from PTB, serving the purpose of model performance assessment during training. The remaining 108 videos, extracted from Tracklam, Track3D, DET [75], People [68], and Scene [33] constitute the test set, designed to evaluate the performance of the models. The videos derived from the DET, People, and Scene datasets are fully preserved within the test set, facilitating a comprehensive evaluation of various SOD models and corroborating their ability to generalize across different scenarios. As presented in Table 2, we report the benchmarking results for each subset and provide an overall measurement.

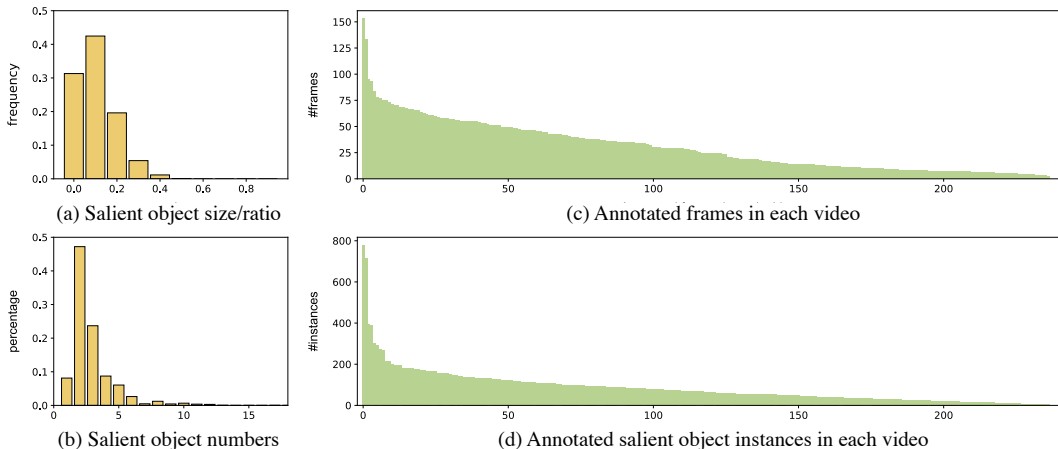

Figure 3: Statistics of the proposed DViSal dataset in terms of (a) salient object size, (b) salient object numbers at image level, as well as (c) the number of annotated frames and (d) instances at video level.

## 3.2 Dataset Statistics

**Comparison with Existing Datasets.** Table 1 presents a comparative overview of popular SOD datasets and our proposed DViSal dataset. As observed, our DViSal dataset offers several notable advantages over existing datasets. First, DViSal integrates more rich data types in both RGB and depth views and video sequences. This stands in contrast to current datasets such as COME [79] and DAVSOD [15], which either provide single pairs of RGB and depth images or solely RGB video sequences as input. Second, the DViSal furnishes a more comprehensive set of annotations. Compared to typical SOD datasets that often involve only one type of object-level annotation [31, 36, 60, 62, 74], the diverse annotation sets in our DViSal dataset enable the exploration of additional domains such as instance-level or weakly-supervised DVSOD. This expanded capacity for research could foster a wider spectrum of research opportunities and catalyze the development of more sophisticated algorithms. Additionally, our dataset covers a wide range of scenarios and contains many challenging cases, such as cluttered background in Fig. 1 and multi-object scenario in Fig. 2. The diversity will be a valuable asset in assessing the scalability of saliency models.

**Statistical Analysis.** Fig. 3 provides the statistical results of our DViSal dataset from several dimensions. (1) *Size of salient objects*: The sizes of salient objects in the dataset significantly impact the performance of saliency detection, as smaller objects are generally more challenging to detect. In our analysis, we define the object size (OS) as the proportion of salient pixels to the total number of pixels in the annotated frames. The results in Fig. 3 (a) reveal that the DViSal dataset exhibits a relatively small proportion of images with large objects, and the size of salient objects varies widely

within a broad range of 0.02% to 54.55%. Moreover, the mean OS score is 14.65%, which is notably appealing compared to the score of 23.16% in current popular RGB-D datasets as reported in [29]. (2) *Number of salient objects*: Fig. 3 (b) analyzes the distribution of the number of salient objects within the images. It is observed that images with a single salient object constitute only 8% of our dataset, whereas these with multiple salient objects (*i.e.*, the number of salient objects $\geq 3$) constitute a significant 45.26%, surpassing the majority of existing RGB-D SOD and VSOD datasets as reported in [29, 15]. These findings highlight not only the diversity and challenge presented in our dataset but also underscore its potential for advancing saliency detection research. (3) *Video-level statistics*: we give video-level analysis on the distribution of labeled frames and the number of salient instances in the videos, with results depicted in Figs. 3 (c) and (d), respectively. These results further showcase the versatility of the proposed DViSal dataset.

## 4 The DVSOD Benchmark

### 4.1 DVSOD Baseline

To date, various network architectures [28, 83] have been developed for the tasks of RGB-D SOD and VSOD. In the former task, advanced feature fusion techniques are adopted to fuse features extracted from RGB-D images based on two-stream encoders. The latter task focuses more on exploiting temporal associations in video sequence. Based on existing techniques in RGB-D SOD [27] and VSOD [57], we assemble a straightforward DVSOD baseline, aiming at examining the advantages of utilizing depth and temporal information in improving detection accuracy & robustness.

Fig. 4 shows an overview of the proposed DVSOD baseline, which consists of four steps: 1) feature extraction; 2) fusion of RGB-D multimodal information; 3) video temporal aggregation; and 4) prediction of salient objects. Specifically, the input is a RGB-D video clip that contains a Query pair of RGB and depth images at current frame $t$, and $L$ Memory pairs at past frames. We first feed these image pairs into two-stream encoders to extract RGB and depth features, respectively. Then, a multimodal fusion module, *i.e.*, cross reference modules (CRM) [27], is adopted to unify the information from two modalities at each network layer. The features across multiple layers are then decoded by the segmentation decoder [73],

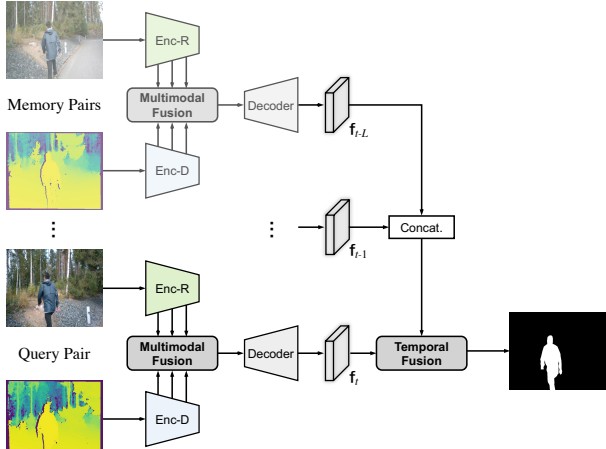

Figure 4: An overview of the DVSOD baseline.

resulting in a series of fused multimodal features. We represent them as $\{\mathbf{f}_d \in \mathbb{R}^{H \times W \times D}\}_{d \in U}$, where $H \times W$ represents the spatial size, $D$ is the channel dimension, and $d$ represents the time subscript of a certain frame in the set of $U = \{t - L, \dots, t - 1, t\}$. To incorporate the temporal cues, we exploit a basic temporal fusion module, *i.e.*, the space-time memory read block [57], to furnish the Query features by engaging the rich features of Memory frames. Finally, we apply a $1 \times 1$ convolutional layer on the memory-augmented feature to predict the saliency map, which is supervised by the ground-truth saliency map using the conventional binary cross entropy (BCE) loss. More details to DVSOD baseline are described in the supplementary materials.

**Implementation Details.** The framework is implemented with PyTorch and trained using a NVIDIA RTX A6000 GPU. We utilize the CPD decoder [73] in conjunction with the ResNet-50 encoder [21] as the base network. For the depth stream, we generate 3-channel depth maps by repeating the 1-channel depth maps. Each image is uniformly resized to $320 \times 320$, and we perform random horizontal flipping and cropping to avoid potential over-fitting. During model training, the learning rate is set to 1e-4, and Adam optimizer is adopted with mini-batch of 2. In the inference stage, the proposed baseline predicts saliency maps in an end-to-end manner and no post-processing procedure (*e.g.*, CRF [22]) is applied in this work.

## 4.2 Evaluation Metrics

We adopt four widely-used metrics to evaluate the performance of saliency models, including mean absolute error (MAE or $M$) [3], F-measure ($F_\beta$) [1], S-measure ($S_\alpha$) [12] and E-measure ($E_\xi$) [13]. *The lower the MAE, the better. For other metrics, the higher score is better.* Concretely, F-measure is an overall performance measurement and is computed by the weighted harmonic mean of the precision and recall:

$$F_\beta = \frac{(1 + \beta^2) \times Precision \times Recall}{\beta^2 \times Precision + Recall}, \tag{1}$$

where $\beta^2$ is set to 0.3 as suggested in [1]. $M$ represents the average absolute difference between the saliency map and ground truth. It is used to calculate how similar a normalized saliency maps $\mathcal{S} \in [0, 1]^{W \times H}$ is compared to the ground truth $\mathcal{G} \in \{0, 1\}^{W \times H}$:

$$M = \frac{1}{W \times H} \sum_{x=1}^{W} \sum_{y=1}^{H} |\mathcal{S}(x, y) - \mathcal{G}(x, y)|, \tag{2}$$

where $W$ and $H$ denote the width and height of $\mathcal{S}$, respectively. Structural measure (S-measure) evaluates the structural similarity between the predicted saliency maps and the binary ground truths. S-measure (denoted as $S_\alpha$) contains two terms, $S_o$ and $S_r$, referring to object-aware and region-aware structural similarities, respectively:

$$S_\alpha = \lambda * S_o + (1 - \lambda) * S_r \tag{3}$$

where $\lambda$ is the balance parameter and is set to 0.5 as in [12]. Enhanced-alignment measure ($E_\xi$) considers the global means of the image and local pixel matching simultaneously.

$$E_\xi = \frac{1}{W \times H} \sum_{i=1}^{W} \sum_{j=1}^{H} \phi_s(i, j), \tag{4}$$

where $\phi_s(\cdot)$ is the enhanced alignment matrix, which reflects the correlation between $\mathcal{S}$ and $\mathcal{G}$ after subtracting their global means, respectively.

## 4.3 Benchmark Results

In this section, we benchmark the DVSOD task by conducting a range of experiments using the DViSal dataset with 11 popular state-of-the-art SOD methods. These methods include four *RGB-image-based SOD models* (PoolNet [43], BASNet [65], CPD [73], and F3Net [72]), five *RGB-D SOD models* (DMRA [62], CoNet [28], BBSNet [16], RD3D [7] and SPNet [95]), two *VSOD models* (PCSA [19] and UGPL [63]), and our *DVSOD baseline*. We obtained results from these methods on the new dataset by reproducing them with their publicly available codes and default setups.

**Quantitative Results.** In DVSOD, one important expectation compared to the RGB-based model is whether properly utilizing RGB-D and temporal features improves the per-frame segmentation accuracy. To verify it, we first conduct three comparative experiments (last three columns of Table 2) to evaluate the influence of each data type. The basic model, CPD [73], which is trained with just RGB images, achieves an overall Mean Absolute Error (MAE) score of 13.2%. When we incorporate a depth component (CPD + Dep), the error significantly decreases by 1.1%. As we employ multimodal RGB-D data and temporal video data jointly (CPD + DepVid), the performance further improves to 11.3%. These results consistently highlight the advantages of using 3D layout data and temporal contexts to locate salient objects. Considering that DVSOD is relatively new and its development is still at an early stage, we additionally present the segmentation results of related SOD/RGB-D SOD/VSOD models in Table 2, to provide observational reference for our readers.

**Qualitative Results.** Fig. 5 visualizes the saliency predictions from different saliency models on the DViSal dataset. We find that the model trained with both RGB-D data and video data generates saliency maps that closely match the ground truths. For example, the DVSOD baseline effectively identifies entire objects even in challenging low-light situations as displayed in Fig. 5 where other methods struggle. We attribute this success to the combined benefits of multimodal and temporal contexts. In addition, our DViSal dataset's diverse scenarios in Figs. 1-2&5 demonstrate its versatility to provide a sufficiently realistic benchmark in this field.

Table 2: Benchmarking results of related SOD approaches tested on the newly-proposed DViSal dataset. We report the numerical evaluation for each test subset and provide an overall measurement. ↑ & ↓ denote larger and smaller is better, respectively. The notations † and ‡ refer to RGB-D SOD and VSOD models, respectively. The best performing model is highlighted in **bold**.

| * | Model | PoolNet [43] | BASNet [65] | F3Net [72] | DMRA [62]† | CoNet [28]† | BBSNet [16]† | RD3D [7]† | SPNet [95]† | PCSA [19]‡ | UGPL [63]‡ | CPD [73] | CPD +Dep† | CPD +DepVid†‡ |
|---|---|---|---|---|---|---|---|---|---|---|---|---|---|---|
| DET | $E_\xi\uparrow$ | .762 | .761 | .766 | .768 | .775 | .784 | .786 | **.788** | .771 | .781 | .761 | .781 | .782 |
| | $S_\alpha\uparrow$ | .666 | .672 | .688 | .678 | .696 | .692 | .695 | .686 | .680 | .688 | .677 | .683 | **.705** |
| | $F_\beta\uparrow$ | 520 | .528 | .543 | .532 | .549 | .563 | .562 | .565 | .535 | .552 | .532 | .554 | **.568** |
| | $M\downarrow$ | .143 | .142 | .146 | .135 | .139 | .134 | .127 | **.126** | .134 | .136 | .142 | .139 | .129 |
| Scene | $E_\xi\uparrow$ | .769 | .826 | .905 | .838 | .830 | **.906** | .852 | .888 | .837 | .868 | .804 | .880 | .881 |
| | $S_\alpha\uparrow$ | .646 | .682 | .813 | .738 | .772 | .802 | .722 | .728 | .738 | .768 | .719 | .786 | **.825** |
| | $F_\beta\uparrow$ | .441 | .514 | .751 | .571 | .652 | **.759** | .575 | .749 | .569 | .721 | .535 | .733 | .722 |
| | $M\downarrow$ | .093 | .075 | **.043** | .065 | .062 | **.043** | .065 | .049 | .067 | .050 | .078 | .048 | **.043** |
| People | $E_\xi\uparrow$ | .472 | .467 | .517 | .474 | .488 | .528 | .515 | .574 | .478 | .581 | .409 | .572 | **.584** |
| | $S_\alpha\uparrow$ | .492 | .476 | .510 | .529 | .502 | .553 | .551 | .532 | .541 | .550 | .463 | .511 | **.558** |
| | $F_\beta\uparrow$ | .162 | .114 | .160 | .192 | .138 | .178 | .198 | .192 | .192 | **.224** | .106 | .169 | .207 |
| | $M\downarrow$ | .153 | .146 | .119 | .139 | .128 | .086 | .103 | .092 | .128 | **.078** | .254 | .098 | .085 |
| Track3D | $E_\xi\uparrow$ | .657 | .600 | .780 | .634 | .658 | .839 | .655 | **.860** | .638 | .838 | .612 | .831 | .822 |
| | $S_\alpha\uparrow$ | .590 | .589 | **.779** | .629 | .655 | .724 | .633 | .667 | .629 | .693 | .600 | .726 | .751 |
| | $F_\beta\uparrow$ | .301 | .290 | .567 | .343 | .423 | .664 | .369 | **.682** | .342 | .657 | .322 | .621 | .609 |
| | $M\downarrow$ | .127 | .145 | .087 | .118 | .101 | .080 | .117 | .083 | .116 | .080 | .134 | .080 | **.073** |
| Tracklam | $E_\xi\uparrow$ | .815 | .798 | .913 | .828 | .896 | .923 | .809 | .889 | .833 | .924 | .787 | .915 | **.925** |
| | $S_\alpha\uparrow$ | .701 | .682 | .769 | .717 | .761 | .793 | .697 | .739 | .717 | .785 | .702 | .771 | **.804** |
| | $F_\beta\uparrow$ | .622 | .595 | .779 | .652 | .751 | **.809** | .628 | .759 | .659 | .803 | .608 | .792 | .795 |
| | $M\downarrow$ | .122 | .133 | .076 | .114 | .082 | .068 | .127 | .084 | .112 | **.063** | .133 | .069 | **.063** |
| PTB | $E_\xi\uparrow$ | .908 | .880 | .909 | .893 | .913 | .924 | .925 | .911 | .894 | .919 | .888 | .906 | **.927** |
| | $S_\alpha\uparrow$ | .793 | .791 | .826 | .810 | .824 | .834 | .811 | .793 | .808 | .832 | .816 | .819 | **.842** |
| | $F_\beta\uparrow$ | .826 | .710 | .795 | .753 | .808 | .821 | **.843** | .835 | .754 | .837 | .745 | .816 | .833 |
| | $M\downarrow$ | .062 | .067 | .057 | .060 | .056 | .054 | .053 | .060 | .060 | .052 | .058 | .052 | **.049** |
| Overall | $E_\xi\uparrow$ | .774 | .774 | .795 | .783 | .794 | **.810** | .799 | **.810** | .786 | .806 | .772 | .805 | .807 |
| | $S_\alpha\uparrow$ | .673 | .679 | .711 | .692 | .713 | .715 | .703 | .698 | .694 | .709 | .689 | .705 | **.729** |
| | $F_\beta\uparrow$ | .537 | .540 | .589 | .555 | .585 | .609 | .582 | .608 | .557 | .598 | .548 | .599 | **.610** |
| | $M\downarrow$ | .133 | .132 | .128 | .125 | .124 | .118 | .118 | **.113** | .123 | .119 | .132 | .121 | **.113** |

**Diagnostic Analysis.** Here we discuss the impact of memory size $\mathcal{M}$ and sample rate $S$ for memory frame selection. As shown in Table 3, adding memory frames consistently improves $F_\beta$ scores compared to the single-frame model (*i.e.*, $\mathcal{M} = 0$). When using more memory frames (*i.e.*, $\mathcal{M} = 3$), we see a clear performance increase (*i.e.*, 59.9%→61.0%). Increasing the $\mathcal{M}$ further beyond 3 gives marginal returns in performance. Hence, we set $\mathcal{M} = 3$ as a balance between accuracy and memory usage. We then tested different sample rates while fixing memory size $\mathcal{M} = 3$. It is discovered in Table 4 that the best performance is achieved with a moderate sample rate $S = 3$. Thus, we set both $\mathcal{M}$ and $S$ to 3, effectively using past video frames without retaining too much outdated information.

Table 3: Ablation on the impact of memory size using F-measure and MAE.

| * | $\mathcal{M}=0$ | $\mathcal{M}=1$ | $\mathcal{M}=2$ | $\mathcal{M}=3$ | $\mathcal{M}=4$ |
|---|---|---|---|---|---|
| $F_\beta\uparrow$ | 0.599 | 0.603 | 0.607 | 0.610 | 0.608 |
| $M\downarrow$ | 0.121 | 0.118 | 0.114 | 0.113 | 0.113 |

Table 4: Ablation on the impact of sample rate using F-measure and MAE.

| * | $S=1$ | $S=2$ | $S=3$ | $S=4$ |
|---|---|---|---|---|
| $F_\beta\uparrow$ | 0.605 | 0.608 | 0.610 | 0.605 |
| $M\downarrow$ | 0.117 | 0.114 | 0.113 | 0.114 |

## 5 Discussion and Outlook

In this section, we discuss several potential research problems on the emerging *DVSOD* task, and provide some feasible solutions for reference. Hopefully this could encourage more inspirations and contributions to this community. They are summarized as follows:

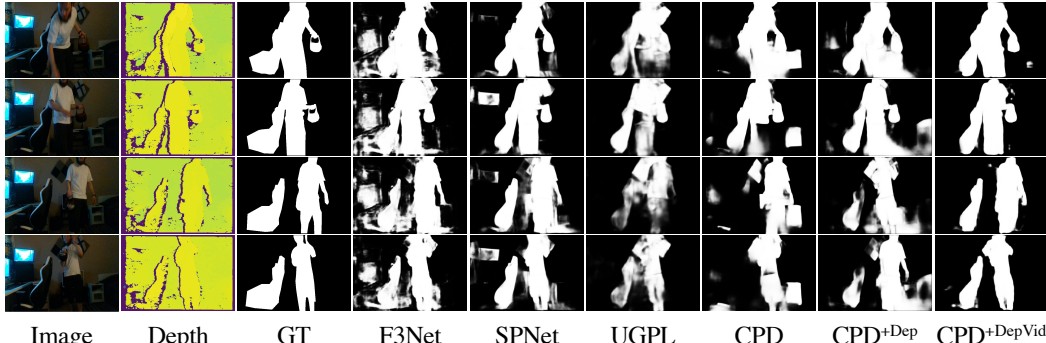

| Image | Depth | GT | F3Net | SPNet | UGPL | CPD | CPD$^{+\mathrm{Dep}}$ | CPD$^{+\mathrm{DepVid}}$ |

Figure 5: Visual results of different SOD models. The 'CPD$^{+\mathrm{DepVid}}$' stands for the DVSOD baseline in Sec. 4.1 that integrates both multimodal RGB-D cues and temporal contexts.

**(1)** *Accuracy*: The research of DVSOD is still in its initial stage and DVSOD research has room for improvement, particularly regarding the accuracy of the DVSOD models. There's potential to enhance the accuracy of DVSOD models by incorporating ideas from the established fields of RGB, RGB-D, and Video SODs. For instance, we could integrate the multi-scale learning techniques [6, 43, 92, 88] into cross-modal and cross-frame fusion to improve the model's contextual representation. Moreover, we could introduce extra edge signals [65, 28] to aid the model in capturing object boundary details. In addition, it could be beneficial to explore more sophisticated fusion techniques [16, 95, 25] to promote effective interaction between multimodal and temporal information.

**(2)** *Efficiency*: Although the engagement of RGB-D videos brings significant improvement, it introduces additional model parameters. More lightweight strategies can be explored to improve efficiency. For instance, we can develop lightweight operations such as depthwise separable convolution [10] or neural architecture search techniques [84], to advance feature extractors. On the other hand, we could explore knowledge distillation schemes [64, 54] to transfer depth structure knowledge to the RGB stream, thereby reducing the heavy overhead of the depth encoder. To evaluate the impact of lightweight strategy, we conduct an exploratory experiment by replacing the conventional convolution in our network encoder with the more efficient depth-wise separable convolution. Our experiments show a reduction in network parameters from 97.3M to 58.1M and memory usage from 17.15G to 16.21G, along with an increase in inference speed from 16.7 FPS to 21.5 FPS. However, this change leads to a drop in performance, with the F-measure falling from 0.610 to 0.575. These findings underscore the potential of efficiency-enhancing strategies, and also highlight the need to balance accuracy with efficiency in saliency model design.

**(3)** *Temporal Modeling*: In this work, we conduct a preliminary investigation on the benefits of RGB-D videos for saliency detection by incorporating an well-designed temporal aggregation scheme [57] that focuses on modeling long-term dependencies. As a potential extension, we could consider introducing optical flow to capture short-term temporal cues. However, it's important to note that directly integrating optical flow could increase training complexity and potentially introduce noise, especially in complex scenes. Meanwhile, it is wroth exploring the saliency change problem (*e.g.*, the salient object changing from a person to a table) when referring to temporal information of previous frames. In the DVSOD baseline, we attempt to address it through engaging the attention mechanism within [57], which selectively discounts less relevant information and fetches the most relevant features. We encourage further investigation on this matter, and believe there is a significant opportunity for the development of more effective temporal modeling schemes that can address these challenges.

**(4)** *Dataset*: As the first dataset proposed for DVSOD, there is still considerable room for improving the various aspects of the dataset such as quality, quantity, and diversity by involving a broader range of scenarios of extreme lighting conditions, as well as various types of occlusions. Furthermore, annotating such large-scale RGB-D video dataset at pixel-level is very costly in terms of both time and effort. We may explore the recent advances of photorealistic rendering techniques [11, 20] to simulate data or supplement annotations to help address challenges within the dataset.

**(5)** *Evaluation Metrics*: Evaluation metrics are crucial for model training, testing, and benchmarking. However, the saliency detection community largely relies on classic metrics such as MAE and F-measure, which are not designed to assess sequential DVSOD tasks specifically. Meanwhile, the Temporal Coherence (TC) metric, frequently used in evaluating RGB video, may not accurately

depict DVSOD model performance due to the complex scenes within the DViSal benchmark, such as low light or cluttered backgrounds. Thus, how to design appropriate metrics specifically tailored for DVSOD remains an open challenge.

**(6) *Instance-level Extension***: As revealed in [18], instance-level extension is capable of recognizing individual instances of salient objects. It is more challenging as it requires detailed parsing within the detected salient regions, and holds significant importance for practical applications. Hence exploring the research on RGB-D video salient instance detection emerges as a promising future direction.

**(7) *Weakly-supervised Learning***: The acquirement of precise per-pixel labels is laborious and time-consuming. As an alternative, training DVSOD models with weak supervisions [80] (*e.g.*, bounding box, scribble) is an appealing research topic, which can avoid heavy annotation costs.

## 6  Conclusion

In this work, we present the DViSal dataset to spur research in RGB-D video salient object detection (DVSOD), a field yet to be well explored. This dataset, composed of 237 diverse RGB-D videos and comprehensive annotations, will significantly expand research avenues. Empirical benchmarking experiments demonstrate the effectiveness of multimodal video input in enhancing salient object detection. We highlight some promising future research directions and make the dataset and results publicly available to accelerate progress in this field.

**Acknowledgements.** This work was done jointly by UA and SRA. This research was supported by the CFI-JELF, Mitacs, University of Alberta Start-up grant, NSERC Discovery (RGPIN-2019-04575) grants.

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
