# Supplementary Material:
# DVSOD: RGB-D Video Salient Object Detection

**Jingjing Li[1], Wei Ji[1,2*], Size Wang[1], Wenbo Li[2], Li Cheng[1]**
[1]University of Alberta, Canada
[2]Samsung Research America AI Center, USA
{jingjin1, wji3, size1, lcheng5}@ualberta.ca, wenbo.li1@samsung.com

In this supplementary material, we provide further elaboration on the DVSOD benchmark and showcase additional results, covering the following aspects.

- Public URLs to DVSOD Benchmark
- More Statistics to DViSal Dataset
- Source Codes to Benchmarking Models
- More Details to DVSOD Baseline
- More Experimental Results
- Additional Analysis
- Maintenance Plan
- Broader Impact
- Datasheet

## A  Appendix

### A.1  Public URLs to DVSOD Benchmark

The reproducibility of DVSOD benchmark is guaranteed by a number of public resources, listed below. This allows easy access and will facilitate replication of the results in our work and future works as well. Our team also promises to maintain the platform and support further developments based on the community feedback. This work is licensed under the GNU General Public License v3.0 (GPL-3.0 license).

- **Project Website**: `https://dvsod.github.io/`
- **DViSal Dataset**: `https://github.com/DVSOD/DVSOD-DViSal`
- **Benchmark Results**: `https://github.com/DVSOD/DVSOD-DViSal`
- **Baseline Code**: `https://github.com/DVSOD/DVSOD-Baseline`
- **Evaluation Tool**: `https://github.com/DVSOD/DVSOD-Evaluation`

### A.2  More Statistics to DViSal Dataset

Table 1 shows more statistical results of the proposed DViSal dataset. Based on the sourced repositories, we detail the total number of videos, the number of RGB-D image pairs, durations of the videos, the number of annotated frames (GTs), the number of salient object instances, and the dataset splits.

37th Conference on Neural Information Processing Systems (NeurIPS 2023) Track on Datasets and Benchmarks.

Table 1: More statistical results of the proposed DViSal dataset.

| Dataset | #Video | #RGB-D Pairs | Duration(s) | #GTs | #Instances | Video Splits |
|---------|--------|--------------|-------------|------|------------|--------------|
| CDTB [12] | 71 | 79,503 | 3,180 | 3,202 | 7,600 | only train |
| PTB [17] | 53 | 12,029 | 481 | 507 | 1,650 | train-27/val-26 |
| Track3D [24] | 5 | 668 | 27 | 29 | 92 | train-2/test-3 |
| Tracklam [1] | 21 | 7,034 | 281 | 291 | 1,324 | train-3/test-18 |
| People [18] | 1 | 224 | 9 | 9 | 33 | only test |
| Scene [8] | 10 | 6,784 | 271 | 275 | 1,102 | only test |
| DET [23] | 76 | 69,200 | 2,768 | 2,804 | 8,425 | only test |

Table 2: Source codes of benchmarking models involved in the paper.

| Method | URL |
|--------|-----|
| PoolNet [10] | https://github.com/backseason/PoolNet |
| BASNet [16] | https://github.com/xuebinqin/BASNet |
| CPD [22] | https://github.com/wuzhe71/CPD |
| F3Net [20] | https://github.com/weijun88/F3Net |
| DMRA [14] | https://github.com/jiwei0921/DMRA |
| CoNet [7] | https://github.com/jiwei0921/CoNet |
| BBSNet [3] | https://github.com/zyjwuyan/BBS-Net |
| RD3D [2] | https://github.com/PolynomialQian/RD3D |
| SPNet [27] | https://github.com/taozh2017/SPNet |
| PCSA [4] | https://github.com/guyuchao/PyramidCSA |
| UGPL [15] | https://github.com/Lanezzz/UGPL |
| BTS-Net [26] | https://github.com/zwbx/BTS-Net |
| VST [11] | https://github.com/nnizhang/VST |
| SPSN [9] | https://github.com/Hydragon516/SPSN |
| DVSOD Baseline (**this work**) | https://github.com/DVSOD/DVSOD-Baseline |

## A.3 Source Codes to Benchmarking Models

Table 2 summarizes the source codes of benchmarking models involved in the main paper.

## A.4 More Details to DVSOD Baseline

In this section, we provide more details about the DVSOD baseline. As described in the main paper, we constructed a basic DVSOD baseline model leveraging methodologies from RGB-D SOD (*i.e.*, CRM [6]) and VSOD (*i.e.*, STM [13]), which are applied for Multimodal Fusion and Temporal Fusion, respectively. Herein, we delve into the foundational principles of these two modules.

**Cross Reference Module** [6] is an efficient fusion module that combines features from both RGB and depth modalities. Generally, features extracted from the RGB channel hold a wealth of semantic and textural information, while depth channel features carry distinguishing scene layout cues. These cues are complementary to the RGB features. CRM's purpose is to extract and blend the most distinctive channels (*i.e.*, feature detectors[21, 25]) among depth and RGB features, thereby producing more informative feature sets.

In detail, we take two input features: $F_i^{RGB}$ and $F_i^{Depth}$, generated by the $i^{th}$ convolutional block of the RGB stream and depth stream, respectively. CRM first utilizes a global average pooling (GAP) process to gather global statistics from the RGB and depth views. Subsequently, the feature vectors are individually passed through a fully connected layer (FC) and a softmax activation function $\delta(\cdot)$ to acquire the channel attention vectors $Att_i^{RGB}$ and $Att_i^{Depth}$ which highlight the significance of RGB and depth features respectively. These attention vectors are applied on the input feature via channel-wise multiplication, causing the CRM to selectively concentrate on important features and mute unnecessary ones for scene understanding. Furthermore, the attention vectors $Att_i^{RGB}$ and $Att_i^{Depth}$ are amalgamated by a maximum function to retain useful feature channels from both streams. They are then subjected to a normalization operation $\mathcal{N}(\cdot)$ to normalize the output in the range 0 to 1, creating a cross-referenced channel attention vector $Att_i^{CR}$. Based on $Att_i^{CR}$, the

enhanced features $\tilde{F}_i^{RGB}$ and $\tilde{F}_i^{Depth}$ can be obtained by summing the $\dot{F}_i^{RGB}$ and $\dot{F}_i^{Depth}$ with the $Att_i^{CR}$ enhanced features. The enhanced features from the RGB branch and depth branch are further concatenated and fed to the $1 \times 1$ convolutional layer to generate the cross-modal fused feature $\mathcal{F}_i$.

**Space-Time Memory Read Block** [13] is a novel DNN system based on the memory network [19, 5] that calculates the spatio-temporal attention for every pixel across multiple video frames for each pixel in the query image. This helps in determining whether the pixel belongs to a foreground object.

In more detail, during video processing, previous frames are considered as memory frames, and the current frame is viewed as the query frame. Both memory and query frames are initially transformed into pairs of key and value maps. In the memory read operation, soft weights are calculated by assessing the similarities between all pixels of the query key map and the memory key map. This similarity matching is conducted in a non-local manner, associating every space-time location in the memory key map with every spatial location in the query key map. The value of the memory is then retrieved by a weighted summation with the soft weights and is concatenated with the query value. With the help of the STM, our DVSOD baseline can effectively leverage temporal contexts to help locate the salient object in current frame.

Table 3: Quantitative results of several state-of-the-art RGB-D SOD methods tested on the newly-proposed DViSal dataset.

| * | DET | | | | Scene | | | | People | | | | Overall | |
|---|---|---|---|---|---|---|---|---|---|---|---|---|---|---|
| | $E_\xi \uparrow$ | $S_\alpha \uparrow$ | $F_\beta \uparrow$ | $M \downarrow$ | $E_\xi \uparrow$ | $S_\alpha \uparrow$ | $F_\beta \uparrow$ | $M \downarrow$ | $E_\xi \uparrow$ | $S_\alpha \uparrow$ | $F_\beta \uparrow$ | $M \downarrow$ | $E_\xi \uparrow$ | $S_\alpha \uparrow$ |
| BTS-Net [26] | 0.781 | 0.688 | 0.559 | 0.133 | 0.854 | 0.710 | 0.667 | 0.059 | 0.579 | 0.542 | 0.180 | 0.087 | 0.802 | 0.700 |
| SPSN [9] | 0.782 | 0.690 | 0.557 | 0.136 | 0.856 | 0.824 | 0.684 | 0.053 | 0.544 | 0.545 | 0.189 | 0.099 | 0.804 | 0.715 |
| VST-RGBD [11] | 0.780 | 0.694 | 0.565 | 0.134 | 0.868 | 0.804 | 0.717 | 0.049 | 0.486 | 0.492 | 0.146 | 0.140 | 0.807 | 0.718 |

| * | Track3D | | | | Tracklam | | | | PTB | | | | Overall | |
|---|---|---|---|---|---|---|---|---|---|---|---|---|---|---|
| | $E_\xi \uparrow$ | $S_\alpha \uparrow$ | $F_\beta \uparrow$ | $M \downarrow$ | $E_\xi \uparrow$ | $S_\alpha \uparrow$ | $F_\beta \uparrow$ | $M \downarrow$ | $E_\xi \uparrow$ | $S_\alpha \uparrow$ | $F_\beta \uparrow$ | $M \downarrow$ | $F_\beta \uparrow$ | $M \downarrow$ |
| BTS-Net [26] | 0.765 | 0.710 | 0.539 | 0.091 | 0.882 | 0.747 | 0.743 | 0.085 | 0.892 | 0.804 | 0.794 | 0.061 | 0.593 | 0.119 |
| SPSN [9] | 0.761 | 0.743 | 0.561 | 0.088 | 0.92 | 0.791 | 0.793 | 0.066 | 0.917 | 0.827 | 0.831 | 0.054 | 0.598 | 0.117 |
| VST-RGBD [11] | 0.787 | 0.716 | 0.588 | 0.085 | 0.921 | 0.801 | 0.786 | 0.065 | 0.917 | 0.838 | 0.818 | 0.052 | 0.606 | 0.118 |

## A.5 More Experimental Results

Table 3 lists the results of three recent cutting-edge RGB-D SOD methods, including BTS-Net [26], VST [11] and SPSN [9], aiming to supplement the benchmark table (Table 2 of main paper) and provide a more comprehensive benchmark.

Fig. 1 provides a set of examples taken from the DViSal dataset, offering an overarching view of the data it contains. A more comprehensive array of visual examples, along with dynamic video visualizations, can be found in our project website.

Fig. 2 displays a selection of sampled frames from six representative video sequences within the DViSal dataset.

- Fig. 2 (a) presents a typical indoor scene featuring a girl showing a book.

- Fig. 2 (b) shows a common scene of several people moving around within a room.

- Fig. 2 (c) showcases a challenging sequence, characterized by fast movement and object occlusion, in which two boys play basketball on an indoor court.

- Fig. 2 (d) features a challenging scene where a football rolls accompanied with camera viewpoint changes.

- Fig. 2 (e) portrays a challenging outdoor scene with multiple objects.

- Fig. 2 (f) illustrates a complex indoor scene with multiple objects under low-light conditions.

Their full demos can be approached at `https://dvsod.github.io/`. As shown in project website, the results obtained from RGBD videos are significantly more appealing when compared to using RGB or RGBD alone as input. This attributes to the superiority of integrating the advantages of complementary depth structure information and temporal associations.

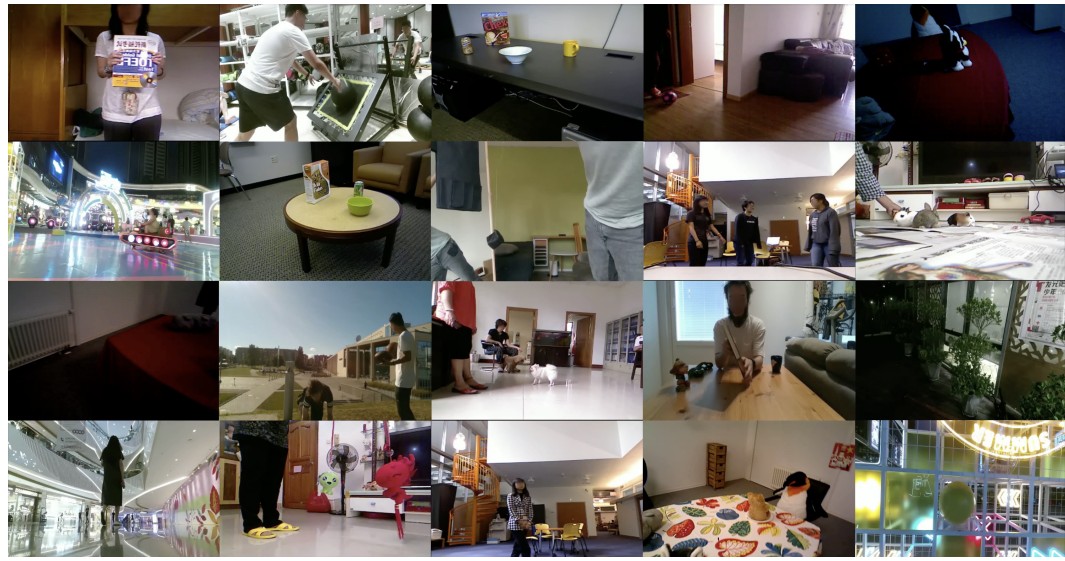

Figure 1: Some examples from the DViSal dataset. For more samples, please visit our project website. The entire dataset is accessible via this link.

## A.6    Additional Analysis

In Section 4.3 (quantitative results) of the main paper, it is observed that the DVSOD baseline performs relatively poorly in Track3D and is inferior to SPNet in terms of F-measure and E-measure. We carefully analyzed the segmentation results of SPNet and DVSOD baseline on Track3D, and conjectured that their disparity in performance is mainly due to the different evaluation criteria of the four evaluation metrics. Specifically, S-measure and MAE are designed to gauge the relative distance between the non-binary saliency maps and the ground truths, while E-measure and F-measure concentrate on the absolute difference between the binary saliency maps and the ground truths.

When evaluated on Track3D dataset, we observed that SPNet's non-binary saliency maps generally produced lower confidence values and highlighted only part of the target objects, which made it performs poorly on S-measures and MAE. When the saliency maps were binarized, these low-confidence predictions were amplified to 0 or 1, and benefit from the adaptive thresholds, some object regions that would otherwise not be significant in the non-binary saliency maps were also extracted. This led to the better E-measure and F-measure scores for SPNet.

Conversely, the DVSOD baseline generated non-binary saliency maps that were notably more confident and encompassed the majority of the object areas, thus yielding superior MAE and S-measure results. However, the binarization process exacerbated some low-density noise that appears at the edges of the object or in the background, making the scores of the E-measure and F-measure not appreciable.

## A.7    Maintenance Plan

The DViSal dataset is hosted on GitHub, and is supported and maintained by the authors. New versions of the DViSal dataset will be shared and announced on our GitHub repository (`https://github.com/DVSOD/DVSOD-DViSal`) if corrections are necessary.

## A.8    Broader Impact

The proposed DViSal dataset offers significant value in enhancing the performance of salient object detection and propelling further research in the field of RGB-D Video Salient Object Detection (DVSOD). Beyond the conventional uses, our dataset introduces potential new applications due to its comprehensive instance-level annotations and weak annotations, which include bounding boxes and scribbles. These varied annotations open avenues for research in instance-level DVSOD and weakly-supervised DVSOD. We envisage that the most proximate impacts of this dataset will be

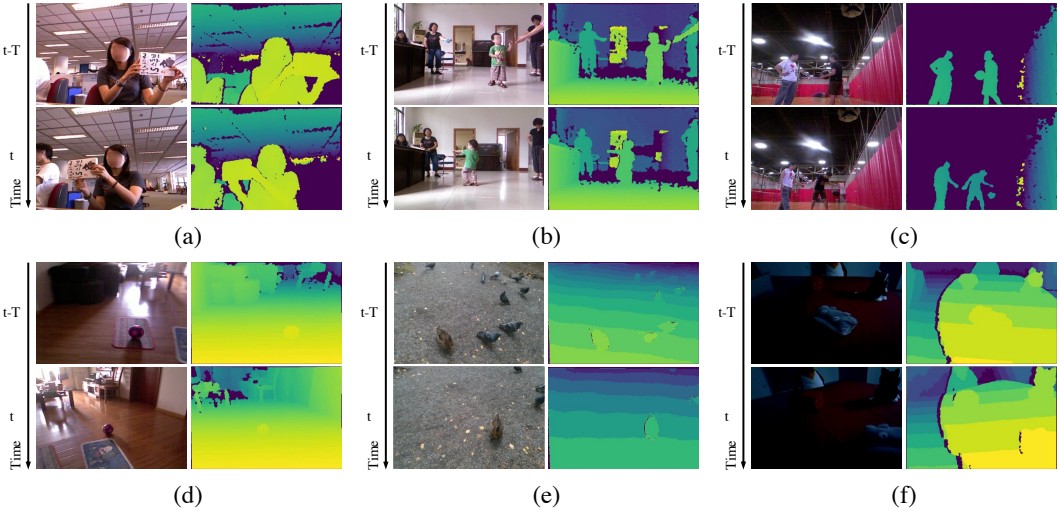

Figure 2: Sample frames from six video sequences.

positive, providing a powerful tool for researchers and developers in the field. Considering the nature of the dataset and the context in which it is likely to be used, we do not foresee any negative societal or ethical implications stemming from its usage.

## B    Datasheet

### B.1    Motivation

- **Q1: For what purpose was the dataset created?** Was there a specific task in mind? Was there a specific gap that needed to be filled? Please provide a description.

  **A1:** The DViSal dataset was established with the specific aim of providing the first publicly accessible RGB-D video salient object detection (DVSOD) dataset. Prior to its creation, there was a noticeable lack of data resources in this particular area, which posed challenges to progress in related research fields. As such, the DViSal dataset was designed to bridge this gap and facilitate advancements in DVSOD and associated studies.

  Beyond its primary function, the creators of the DViSal dataset also envisioned its potential to broaden the scope of research in related fields. The dataset has significant value as a resource for exploring various potential research directions, such as instance-level DVSOD and weakly-supervised DVSOD. Thus, it serves not just as a tool for existing problems but also as a foundation for future investigations and developments.

- **Q2: Who created the dataset (e.g., which team, research group) and on behalf of which entity (e.g., company, institution, organization).**

  **A2:** The DViSal dataset is created by researchers from the University of Alberta and Samsung Research America.

- **Q3: Who funded the creation of the dataset?** If there is an associated grant, please provide the name of the grantor and the grant name and number.

  **A3:** The creation of the DViSal dataset is supported by the CFI-JELF, Mitacs, University of Alberta Start-up grant, NSERC Discovery (RGPIN-2019-04575) grants.

### B.2    Composition

- **Q1: What do the instances that comprise the dataset represent (e.g., documents, photos, people, countries)?** Are there multiple types of instances (e.g., movies, users, and ratings; people and interactions between them; nodes and edges)? Please provide a description.

  **A1:** The DViSal dataset comprises 237 RGB-D videos at a frame rate of 25 f/s, including 175,442 RGB-D pairs in total and 7,117 annotated frames. The dataset has comprehensive annotations, including object and instance level boundary delineations, as well as bounding boxes and scribbles.

- **Q2: How many instances are there in total (of each type, if appropriate)?**
  **A2:** The DViSal dataset encompasses 237 RGB-D videos, collectively consisting of 175,442 RGB-D frame pairs. Among these, there are 7,117 frame pairs that have been extensively annotated for the salient object detection task. In addition to the conventional object-level annotations, the dataset also provides instance-level dense annotations. The total number of instances is 20,226.

- **Q3: Does the dataset contain all possible instances or is it a sample (not necessarily random) of instances from a larger set?** If the dataset is a sample, then what is the larger set? Is the sample representative of the larger set (e.g., geographic coverage)? If so, please describe how this representativeness was validated/verified. If it is not representative of the larger set, please describe why not (e.g., to cover a more diverse range of instances, because instances were withheld or unavailable).
  **A3:** The annotations in the DViSal dataset are made on sparse frames, with a frame being annotated approximately every 25 frames. The practice of sparse annotation primarily aims to alleviate the burden of annotating precise ground truths and prevent redundancy among adjacent frames with similar content and style. Additionally, each annotated frame only contains salient instances that are pertinent to the study of salient object detection.

- **Q4: What data does each instance consist of?** "Raw" data (e.g., unprocessed text or images) or features? In either case, please provide a description.
  **A4:** Every instance in the DViSal dataset is comprised of both raw and annotated data. The raw data is in the form of RGB and depth images, which collectively form a frame pair. These pairs of images capture appearance and depth information for each frame in the dataset, providing a foundation for subsequent analysis. The annotated data, on the other hand, is in the form of comprehensive labels attached to each instance. These labels comprise object-level pixel-wise annotations, instance-level IDs, scribble annotations, and bounding boxes. The annotations provide a detailed understanding of the salient features within the frames, such as the position, shape, and boundaries of objects.

- **Q5: Is there a label or target associated with each instance?** If so, please provide a description.
  **A5:** Each instance in the DViSal dataset is associated with a set of labels. These labels take the form of object-level, instance-level, scribble, and bounding boxes annotations.

- **Q6: Is any information missing from individual instances?** If so, please provide a description, explaining why this information is missing (e.g., because it was unavailable). This does not include intentionally removed information, but might include, e.g., redacted text.
  **A6:** Everything is included. No data is missing.

- **Q7: Are relationships between individual instances made explicit (e.g., users' movie ratings, social network links)?** If so, please describe how these relationships are made explicit.
  **A7:** In the DViSal dataset, relationships between instances are inherently established through the sequence of RGB and depth pairs derived from video sequences. The individual frames within the same video are temporally coherent, meaning that each frame is connected to the next through the continuity of the captured scene. This temporal coherence forms the basis for the relationship between instances. Furthermore, during the annotation process, it is crucial to maintain consistency for adjacent frames. This ensures that the annotated objects and their attributes remain coherent and accurate across the progression of frames, thus preserving the inherent relationships between individual instances in the video sequence.

- **Q8: Are there recommended data splits (e.g., training, development/validation, testing)?** If so, please provide a description of these splits, explaining the rationale behind them.
  **A8:** The entire dataset is partitioned into training, validation and test sets, which consist of 103, 26, and 108 videos, respectively, with 3,560, 200, and 3,357 annotated frames each. To be specific, the training set includes the entire CDTB [12] dataset, which contains 71 video sequences, and an additional 32 randomly selected videos from the PTB [17], Tracklam [1] and Track3D [24] datasets. The validation set is made up of 26 videos sourced from PTB, serving the purpose of model performance assessment during training. The

remaining 108 videos, extracted from Tracklam, Track3D, DET [23], People [18], and Scene [8] constitute the test set, designed to evaluate the performance of the models. The videos derived from the DET, People, and Scene datasets are fully preserved within the test set, facilitating a comprehensive evaluation of various SOD models and corroborating their ability to generalize across different scenarios.

- **Q9: Are there any errors, sources of noise, or redundancies in the dataset?** If so, please provide a description.

  **A9:** Like any dataset, the DViSal dataset is not entirely free from potential sources of noise and errors. These can occur in the form of incorrect labels or fundamental ambiguities in the labeling process. Despite these inherent uncertainties, the team behind the DViSal dataset has implemented multiple quality assurance steps throughout the annotation process. The purpose of these measures is to significantly reduce the incidence of noise and errors, thus ensuring the dataset's reliability and usefulness for research purposes.

- **Q10: Is the dataset self-contained, or does it link to or otherwise rely on external resources (e.g., websites, tweets, other datasets)?** If it links to or relies on external resources, a) are there guarantees that they will exist, and remain constant, over time; b) are there official archival versions of the complete dataset (i.e., including the external resources as they existed at the time the dataset was created); c) are there any restrictions (e.g., licenses, fees) associated with any of the external resources that might apply to a future user? Please provide descriptions of all external resources and any restrictions associated with them, as well as links or other access points, as appropriate.

  **A10:** The DViSal dataset integrates original videos that are gathered from a variety of repositories. These include CDTB [12], People [18], PTB [17], Scene [8], DET [23], Tracklam [1], and Track3D [24]. a) These datasets are made accessible via their respective project websites, which are designed to provide long-term preservation of the data. Hence, the data's continued existence can generally be ensured. b) To circumvent potential issues with consistently extracting frames from videos, we will supply our own copies of the frames that we utilized in our research. c) Certain licensing restrictions are applicable to the original datasets, which are primarily limited to non-commercial usage. It should be noted that these restrictions are also applied to the DViSal dataset.

- **Q11: Does the dataset contain data that might be considered confidential (e.g., data that is protected by legal privilege or by doctor–patient confidentiality, data that includes the content of individuals' non-public communications)?** If so, please provide a description.

  **A11:** No.

- **Q12: Does the dataset contain data that, if viewed directly, might be offensive, insulting, threatening, or might otherwise cause anxiety?** If so, please describe why.

  **A12:** No.

- **Q13: Does the dataset relate to people?** If not, you may skip the remaining questions in this section

  **A13:** Yes, the DViSal dataset is related to people, although not in the sense of containing personal information. The main focus of the dataset, salient object detection, aims to replicate human visual attention behavior - the extraordinary human ability to quickly focus attention on the most important parts of visual scenes. This involves understanding and mimicking how humans naturally perceive and prioritize certain visual elements over others. Furthermore, the annotations for salient objects in the dataset are provided by human annotators, adding another layer of human involvement.

- **Q14: Does the dataset identify any subpopulations (e.g., by age, gender)?** If so, please describe how these subpopulations are identified and provide a description of their respective distributions within the dataset.

  **A14:** No, the DViSal dataset does not identify any subpopulations.

- **Q15: Is it possible to identify individuals (i.e., one or more natural persons), either directly or indirectly (i.e., in combination with other data) from the dataset?** If so, please describe how.

**A15:** While the DViSal dataset does not directly contain personal identifiers, individuals may potentially be identifiable in the original RGB videos. These videos were captured in real-world circumstances and might contain unobscured appearances of people.

- **Q16: Does the dataset contain data that might be considered sensitive in any way (e.g., data that reveals racial or ethnic origins, sexual orientations, religious beliefs, political opinions or union memberships, or locations; financial or health data; biometric or genetic data; forms of government identification, such as social security numbers; criminal history)?** If so, please provide a description.

  **A16:** No, the DViSal dataset does not contain any data that could be considered sensitive according to these criteria. It's primarily focused on salient object detection in visual scenes, and it does not capture or store sensitive personal data.

## B.3 Collection Process

- **Q1: How was the data associated with each instance acquired?** Was the data directly observable (e.g., raw text, movie ratings), reported by subjects (e.g., survey responses), or indirectly inferred/derived from other data (e.g., part-of-speech tags, model-based guesses for age or language)? If the data was reported by subjects or indirectly inferred/derived from other data, was the data validated/verified? If so, please describe how.

  **A1:** The raw data, in the form of videos, was directly observed and collected from several existing repositories including CDTB, People, PTB, Scene, DET, Tracklam, and Track3D. These videos were directly recorded using camera devices. Each video was individually reviewed by the authors who meticulously eliminated any unqualified videos or frames. These could include frames that were blurry, misaligned, or had ambiguous salient objects. The annotation of these RGB-D videos was carried out using the Labelme toolkit. The acquisition and validation of the ground-truth saliency followed the annotation principle utilized in well-accepted SOD datasets.

- **Q2: What mechanisms or procedures were used to collect the data (e.g., hardware apparatuses or sensors, manual human curation, software programs, software APIs)?** How were these mechanisms or procedures validated?

  **A2:** The raw data was collected from multiple existing repositories. The annotation of these RGB-D videos was conducted using the Labelme toolkit, which is widely accepted in the field of SOD.

- **Q3: If the dataset is a sample from a larger set, what was the sampling strategy (e.g., deterministic, probabilistic with specific sampling probabilities)?**

  **A3:** Initially, over 703 RGB-D videos were gathered from the source repositories. We performed a quality assurance strategy, wherein unqualified videos were removed. The final DViSal dataset comprises 237 high-quality RGB-D videos with a total of 175,442 paired frames.

- **Q4: Who was involved in the data collection process (e.g., students, crowdworkers, contractors) and how were they compensated (e.g., how much were crowdworkers paid)?**

  **A4:** The raw data for the DViSal dataset was collected by the authors involved in this work. The authors do not have specific information regarding the individuals who participated in the collection of the source datasets from which our videos were derived. The annotation task was carried out by a group of crowdworkers.

- **Q5: Over what timeframe was the data collected?** Does this timeframe match the creation timeframe of the data associated with the instances (e.g., recent crawl of old news articles)? If not, please describe the timeframe in which the data associated with the instances was created.

  **A5:** The process of collecting raw videos and generating annotations for the DViSal dataset spanned a period of 13 months, specifically from March 2022 to April 2023.

- **Q6: Were any ethical review processes conducted (e.g., by an institutional review board)?** If so, please provide a description of these review processes, including the outcomes, as well as a link or other access point to any supporting documentation.

  **A6:** Not applicable to DViSal. As the DViSal dataset comprises new annotations of pre-existing videos, there were no additional ethical review processes conducted specifically

for this dataset. The ethical considerations for the source datasets, from which the original videos were obtained, were handled by the respective institutions and authors.

- **Q7: Did you collect the data from the individuals in question directly, or obtain it via third parties or other sources (e.g., websites)?**
  **A7:** Not applicable to DViSal.

- **Q8: Were the individuals in question notified about the data collection?** If so, please describe (or show with screenshots or other information) how notice was provided, and provide a link or other access point to, or otherwise reproduce, the exact language of the notification itself.
  **A8:** Not applicable to DViSal. As the dataset consists of pre-existing videos, the participants were aware of the data collection process during the original recording and were given the opportunity to ask questions and withdraw at any time.

- **Q9: Did the individuals in question consent to the collection and use of their data?** If so, please describe (or show with screenshots or other information) how consent was requested and provided, and provide a link or other access point to, or otherwise reproduce, the exact language to which the individuals consented.
  **A9:** Not applicable to DViSal.

- **Q10: If consent was obtained, were the consenting individuals provided with a mechanism to revoke their consent in the future or for certain uses?** If so, please provide a description, as well as a link or other access point to the mechanism (if appropriate).
  **A10:** Not applicable to DViSal.

- **Q11: Has an analysis of the potential impact of the dataset and its use on data subjects (e.g., a data protection impact analysis) been conducted?** If so, please provide a description of this analysis, including the outcomes, as well as a link or other access point to any supporting documentation.
  **A11:** The source datasets, from which the original videos were obtained, were publicly available for research purposes. Since the data in DViSal is anonymous and does not contain personally identifiable information, no additional data protection impact analysis was required for DViSal.

## B.4 Preprocessing/cleaning/labeling

- **Q1: Was any preprocessing/cleaning/labeling of the data done (e.g., discretization or bucketing, tokenization, part-of-speech tagging, SIFT feature extraction, removal of instances, processing of missing values)?** If so, please provide a description. If not, you may skip the remaining questions in this section.
  **A1:** Yes, preprocessing, cleaning, and labeling of the data were conducted as part of the dataset creation process. The main aspects of the preprocessing and cleaning processes involved the removal of unqualified videos or frames. Any instances that were blurry, misaligned, or featured ambiguous salient objects were removed to ensure the quality of the dataset. For labeling, we utilized the Labelme toolkit to annotate the collated RGB-D videos, adhering to the annotation principles utilized in widely-recognized SOD datasets.

- **Q2: Was the "raw" data saved in addition to the preprocessed/cleaned/labeled data (e.g., to support unanticipated future uses)?** If so, please provide a link or other access point to the "raw" data.
  **A2:** No.

- **Q3: Is the software that was used to preprocess/clean/label the data available?** If so, please provide a link or other access point.
  **A3:** Yes, the software used for annotating the dataset, the Labelme toolkit, is publicly available. The link to the software is `https://github.com/wkentaro/labelme`.

## B.5 Uses

- **Q1: Has the dataset been used for any tasks already?** If so, please provide a description.
  **A1:** As the first dataset of its kind, the DViSal dataset is developed specifically for the task of RGB-D Video SOD (DVSOD), pioneering this new field.

- **Q2: Is there a repository that links to any or all papers or systems that use the dataset?** If so, please provide a link or other access point.

  **A2:** No repository exists that links to any papers or systems that use the DViSal dataset.

- **Q3: What (other) tasks could the dataset be used for?**

  **A3:** While initially conceived for RGB-D Video SOD (DVSOD), the DViSal dataset provides an array of diverse annotation sets that extend its utility beyond this primary task. The dataset includes object-level annotations, comprehensive instance-level annotations, and weak annotations such as bounding boxes and scribbles. Consequently, it can be leveraged in a broader spectrum of research areas, including instance-level DVSOD, weakly-supervised DVSOD, and more.

- **Q4: Is there anything about the composition of the dataset or the way it was collected and preprocessed/cleaned/labeled that might impact future uses?** For example, is there anything that a dataset consumer might need to know to avoid uses that could result in unfair treatment of individuals or groups (e.g., stereotyping, quality of service issues) or other risks or harms (e.g., legal risks, financial harms)? If so, please provide a description. Is there anything a dataset consumer could do to mitigate these risks or harms?

  **A4:** Users of the DViSal dataset should be mindful that the raw videos incorporate open-source image frames featuring unblurred objects and people. In order to avoid potential privacy concerns, it is important that this data is handled respectfully and in line with relevant privacy policies.

- **Q5: Are there tasks for which the dataset should not be used?** If so, please provide a description.

  **A5:** No.

## B.6 Distribution

- **Q1: Will the dataset be distributed to third parties outside of the entity (e.g., company, institution, organization) on behalf of which the dataset was created?** If so, please provide a description.

  **A1:** Yes, the DViSal dataset has been made publicly accessible via this link.

- **Q2: How will the dataset will be distributed (e.g., tarball on website, API, GitHub)?** Does the dataset have a digital object identifier (DOI)?

  **A2:** The DViSal dataset can be obtained from the designated GitHub repository found at `https://github.com/DVSOD/DVSOD-DViSal`. Currently, the dataset does not possess a Digital Object Identifier (DOI).

- **Q3: When will the dataset be distributed?**

  **A3:** The DViSal dataset has been accessible to the public via GitHub since June 12, 2023.

- **Q4: Will the dataset be distributed under a copyright or other intellectual property (IP) license, and/or under applicable terms of use (ToU)?** If so, please describe this license and/or ToU, and provide a link or other access point to, or otherwise reproduce, any relevant licensing terms or ToU, as well as any fees associated with these restrictions.

  **A4:** Yes, the DViSal dataset is protected under GNU General Public License v3.0 (GPL-3.0 license). The annotations of DViSal dataset are released for non-commercial research purpose only.

- **Q5: Have any third parties imposed IP-based or other restrictions on the data associated with the instances?** If so, please describe these restrictions, and provide a link or other access point to, or otherwise reproduce, any relevant licensing terms, as well as any fees associated with these restrictions.

  **A5:** No.

- **Q6: Do any export controls or other regulatory restrictions apply to the dataset or to individual instances?** If so, please describe these restrictions, and provide a link or other access point to, or otherwise reproduce, any supporting documentation.

  **A6:** No. The only constraint is the adherence to the non-commercial license.

### B.7 Maintenance

- **Q1: Who will be supporting/hosting/maintaining the dataset?**

  **A1:** The DViSal dataset is hosted on GitHub and supported and maintained by the authors.

- **Q2: How can the owner/curator/manager of the dataset be contacted (e.g., email address)?**

  **A2:** For any inquiries, please feel free to email the corresponding author.

- **Q3: Is there an erratum?** If so, please provide a link or other access point.

  **A3:** Not at the time of release. If there are errata or updates, we will provide them on the dataset website.

- **Q4: Will the dataset be updated (e.g., to correct labeling errors, add new instances, delete instances)?** If so, please describe how often, by whom, and how updates will be communicated to dataset consumers (e.g., mailing list, GitHub)?

  **A4:** New versions of the DViSal dataset will be shared and announced on our GitHub repository (`https://github.com/DVSOD/DVSOD-DViSal`) if corrections are necessary.

- **Q5: If the dataset relates to people, are there applicable limits on the retention of the data associated with the instances (e.g., were the individuals in question told that their data would be retained for a fixed period of time and then deleted)?** If so, please describe these limits and explain how they will be enforced.

  **A5:** Not appliable to DViSal.

- **Q6: Will older versions of the dataset continue to be supported/hosted/maintained?** If so, please describe how. If not, please describe how its obsolescence will be communicated to dataset consumers.

  **A6:** Yes, we will support all versions of our dataset. We will maintain the history of versions via our repository (`https://github.com/DVSOD/DVSOD-DViSal`).

- **Q7: If others want to extend/augment/build on/contribute to the dataset, is there a mechanism for them to do so?** If so, please provide a description. Will these contributions be validated/verified? If so, please describe how. If not, why not? Is there a process for communicating/distributing these contributions to dataset consumers? If so, please provide a description.

  **A7:** Users are free to extend the dataset on their own and create derivative works, so long as they follow the license agreement.