# OpenReview forum: "DVSOD: RGB-D Video Salient Object Detection"
_NeurIPS.cc/2023/Track/Datasets_and_Benchmarks — NeurIPS 2023 Datasets and Benchmarks Poster_

### Official Review · Reviewer_to2U · 2023-07-21
**Review of NeurIPS 2023 Track Datasets and Benchmarks Submission144**

**Rating:** 6
**Confidence:** 4
**Clarity:** Yes, the paper is well organized and …

**Strengths:**

1.This article has a well-structured format, including comparisons of datasets， existing methods' performance and proposed a specialized method for the new task.
2.The website of the dataset is published, and well organized. It includes data, code, evaluation and with samples. All of these contents are clearly describled and  easy to follow.

**Additional Feedback:**

The name of the dataset DViSal appears firstly at the introduction part instead of title or abstract part. I would be better to claim it at title or abstract, so that the readers can find or remember it easier.

**Correctness:**

The claims, the dataset construction, and the evaluation methods are all correct.

**Documentation:**

There are adequate details on the data collection, and a URL of the dataset is provided. It is sufficent to support reproducibility.

**Ethics:**

There are several individuals whose faces are clearly visible, have any measures been taken to blur or pixelate the faces in the video to protect the privacy of the individuals?

**Limitations:**

There are some limitaions are listed in discussion part. But the limitation of the newly introduced dataset is not addressed adequtately.

**Opportunities For Improvement:**

1. The application of the research is limited. Although some devices are  capable of capturing both RGB Video information and depth information simultaneously, there usage is not yet widespread. SOD research on RGB video is more significant relatively.
2. The contribution is limited. The data is gathered from other datasets (add annotation and data refinement)

**Relation To Prior Work:**

Yes, it cleary discussed the difference. In the previous work, the depth information or the temporal information is used for the SOD task, but did not consider them together. This paper is first to take these two types of information together.

**Summary And Contributions:**

The paper introduced a new dataset called DViSal (RGB-D Video Saliency detection), including 237 RGB-D videos, 175,442 RGB-D pairs in total and 7,117 annotated frames with instance-level annotation. And 11 Salient Object Detection(SOD) models are implemented and evaluated on the newly proposed dataset. Also a new baseline method is proposed.

---

> ### Author Response · Authors · 2023-08-23
> **Response to Reviewer to2U**
>
> **Dear Reviewer to2U, we sincerely appreciate the time and effort you spent on reviewing our paper. The comments and suggestions are valuable and encouraging. Here are our detailed replies to your questions.**
>
> ------
>
> ***Question1:*** *The application of the research is limited. Although some devices are capable of capturing both RGB Video information and depth information simultaneously, there usage is not yet widespread. SOD research on RGB video is more significant relatively.*
>
> **Answer1:** Thanks for your valuable comment. We agree with you that, at this juncture, SOD research is predominantly centered around RGB videos. Nonetheless, as you've highlighted, we start to witness quite a few mobile phone models (Samsung Galaxy S20 Ultra, Huawei P30 Pro, iPhone 13 Pro, etc.), Apple iPad Pro, and other consumer electronics gadgets capable of capturing both RGB video information and depth information simultaneously. With the rising trend of adopting 3D sensing capacities, the prevalence of such devices is also expected to surge. By introducing the new benchmark dataset - DViSal, our paper aims to explore the relative new task of learning from the multimodal input of RGB & depth videos.
>
> ------
>
> ***Question2:*** *The contribution is limited. The data is gathered from other datasets (add annotation and data refinement)*
>
> **Answer2:** We greatly appreciate the opportunity to better clarify the contributions of our work.
> First, our work presented a comprehensive dataset encompassing a wide range of scenarios, setting as the first real-world RGB-D video saliency detection dataset. Second, our dataset also stands out with its meticulous annotations - beyond the basic object-level annotations, we offer instance-level annotations as well as weak annotations including bounding boxes and scribbles, paving the way for research on varied annotation tiers. Third, in addition to curate the dataset, we've undertaken a thorough benchmarking of it, validated the pivotal role of multimodal video input in elevating salient object detection, and presented valuable insights that could guide the trajectory of future research in this domain. Even though the base data originate from other datasets, the enrichment through annotation, benchmarking, and insights positions the DViSal dataset as a cornerstone for forthcoming saliency detection studies.
>
> ------
>
> ***Limitations:*** *There are some limitaions are listed in discussion part. But the limitation of the newly introduced dataset is not addressed adequtately.*
>
> **Answer:** We genuinely appreciate your insightful advice and have expanded the discussion on the limitations of the DViSal dataset in Section 5, as suggested.
> First, as the first dataset proposed for DVSOD, there is still considerable room for improving the various aspects of the dataset such as quality, quantity, and diversity by involving a broader range of scenarios of extreme lighting conditions, as well as various types of occlusions. Second, annotating such large-scale RGB-D video dataset at pixel-level is very costly in terms of both time and effort. The reviewer 4mrx's suggestion of employing synthetic data to supplement annotations seems a good alternative to help ease the annotation cost. We sincerely thank you for providing the valuable suggestion to improve the quality of our paper.
>
> ------
>
> ***Ethics:*** *There are several individuals whose faces are clearly visible, have any measures been taken to blur or pixelate the faces in the video to protect the privacy of the individuals?*
>
> **Answer:** We deeply value the reviewer's perspective on this matter. We have blurred the faces in the videos and updated our dataset accordingly. Updates can be viewed on our project website.
>
> ------
>
> ***Additional Feedback:*** *The name of the dataset DViSal appears firstly at the introduction part instead of title or abstract part. I would be better to claim it at title or abstract, so that the readers can find or remember it easier.*
>
> **Answer:** We are sorry that the dataset name appeared too late. We've claimed the dataset name in the abstract section to make it more clear.
>
> ------
>
> **Thanks again for your encouragement and effort on strengthening the quality of our paper.**

---

### Official Review · Reviewer_qgpw · 2023-07-21
**Review of this paper**

**Rating:** 6
**Confidence:** 4
**Clarity:** Yes, this paper is well written and e…

**Strengths:**

1. This paper proposes the first real-world RGB-D video salient object detection dataset, DViSal, making a positive contribution to the underexplored field of RGB-D video salient object detection.
2. The dataset is abundant, consisting of 175,442 pairs of RGB-D images, with annotations available for 7,117 frames and videos. It exhibits strong diversity, covering a wide range of real-world scenes and motion patterns, facilitating in-depth research in this field.


**Additional Feedback:**

No

**Correctness:**

Yes, the claims made in the submission is correct and the dataset is constructed in a sound way. It consists of a combined total of 175,442 RGB-D pairs, while carefully curating 7,117 annotated frames and videos to encompass a wide range of real-life scenarios and motion patterns. The evaluation methods and experiment design are appropriate and performed correctly. The author proposed a simple DVSOD baseline model and compared it with some existing models, using four metrics for Salient object detection (MAE, F-measure, S-measure, E-measure), ultimately proving the effectiveness of multimodal and temporal information in improving model performance.

**Documentation:**

Yes，there is sufficient detail on data collection and organization, availability and maintenance, and ethical and responsible use. For benchmarks,  there is sufficient detail to support reproducibility.

**Ethics:**

The source of the proposed dataset has been indicated, and I don’t think there are any ethical issues.

**Limitations:**

This paper proposes an RGB-D video salient object detection dataset, which fills a research gap in the field and promotes the development of related studies. It has no potential negative societal impact. Furthermore, the authors have effectively addressed the limitations of their work. One suggestion here is to explore further to address the inconsistency between salient object patterns when they differ in depth images and RGB images. For example, an object may become salient in the depth image due to its close proximity, while in the RGB image, another object with brighter colors might be considered salient. Further exploration can provide insights into resolving this inconsistency between the two salient object modes.

**Opportunities For Improvement:**

In order to further improve the quality of the paper, the following are some opportunities for improvement:
1. When the saliency object in the video changes, such as the saliency object changing from a person to a table, the temporal information of the previous few frames mostly contains human features at the beginning of the change. This may have a negative impact on the detection of the table in the current frame, which is worth exploring.
2. In the proposed DVSOD baseline model presented in this paper, both multimodal fusion and temporal fusion utilize methods from other existing studies. It would be preferable if the authors could propose their own more innovative approaches.
3. In the experimental results section, the DVSOD baseline model proposed by the author performs relatively poorly in Track3D and is inferior to SPNet in terms of F-measure and E-measure metrics. The reasons for this phenomenon can be further studied and explained.


**Relation To Prior Work:**

Yes, the paper has clearly discussed how this work differs from previous contributions. Previous work mainly focused on salient object detection in RGB-D images or  RGB videos. The main contribution of this paper is to propose the first real-world dataset for salient object detection in RGB-D videos. Based on this, experiments have shown that multimodal and temporal information are helpful in improving model performance.

**Summary And Contributions:**

This paper introduces a new dataset, DViSal, for studying salient object detection in RGB-D videos. This dataset includes 237 RGB-D videos from different scenes and provides comprehensive annotations, including instance level markers, bounding boxes and scribbles. The paper points out that significant progress has been made in the fields of RGB-D SOD and VSOD, but there is still a research gap in combining multimodal and temporal information. Through benchmark experiments using multiple SOD models on this dataset, the paper confirms the effectiveness of using multimodal video inputs for salient object detection, making positive contributions to the research on salient object detection in RGB-D videos.

---

> ### Author Response · Authors · 2023-08-23
> **Response to Reviewer qgpw  (Part 1/2)**
>
> **Dear Reviewer qgpw, we sincerely thank you for thoroughly reviewing our paper and providing these valuable suggestions. Please find below our detailed response to the comments.**
>
> ------
>
> ***Question1:*** *When the saliency object in the video changes, such as the saliency object changing from a person to a table, the temporal information of the previous few frames mostly contains human features at the beginning of the change. This may have a negative impact on the detection of the table in the current frame, which is worth exploring.*
>
> **Answer1:** Thank you for your insightful suggestion. We concur with you that it is worth exploring the saliency change problem when referring to temporal information of previous frames. In our work, we also realized this problem and attempted to address it through engaging the attention mechanism within the space-time memory read block [47]. When the saliency object changes from a person to a table, the attention mechanism will discount less relevant person cues and fetch the most relevant table features selectively.
> We do believe this issue needs deeper investigation. We have added an extended discussion on this topic in the revised manuscript (Sec. 5), and hope this would facilitate further ideas and development in the community.
>
> ------
>
> ***Question2:*** *In the proposed DVSOD baseline model presented in this paper, both multimodal fusion and temporal fusion utilize methods from other existing studies. It would be preferable if the authors could propose their own more innovative approaches.*
>
> **Answer2:** Thank you for your valuable suggestion. We are actively working on designing our own innovative method for DVSOD, aiming to effectively fuse cross-modal and cross-frame features, bridge the modality gap between two modalities, and optimize model efficiency.
> In the current version, our objective is to lay a solid foundation for the underexplored field of RGB-D video salient object detection. We hope that the released the dataset and benchmark can facilitate future research. Updates will be forthcoming.
>
> ------
>
> ***Question3:*** *In the experimental results section, the DVSOD baseline model proposed by the author performs relatively poorly in Track3D and is inferior to SPNet in terms of F-measure and E-measure metrics. The reasons for this phenomenon can be further studied and explained.*
>
> **Answer3:** Thank you for your thoughtful comments. We carefully analyzed the segmentation results of SPNet and DVSOD baseline on Track3D, and conjectured that their disparity in performance is mainly due to the different evaluation criteria of the four evaluation metrics. Specifically, S-measure and MAE are designed to gauge the relative distance between the non-binary saliency maps and the ground truths, while E-measure and F-measure concentrate on the absolute difference between the binary saliency maps and the ground truths.
>
> When evaluated on Track3D dataset, we observed that SPNet's non-binary saliency maps generally produced lower confidence values and highlighted only part of the target objects, which made it performs poorly on S-measures and MAE. When the saliency maps were binarized, these low-confidence predictions were amplified to 0 or 1, and benefit from the adaptive thresholds, some object regions that would otherwise not be significant in the non-binary saliency maps were also extracted. This led to the better E-measure and F-measure scores for SPNet.
>
> Conversely, the DVSOD baseline generated non-binary saliency maps that were notably more confident and encompassed the majority of the object areas, thus yielding superior MAE and S-measure results. However, the binarization process exacerbated some low-density noise that appears at the edges of the object or in the background, making the scores of the E-measure and F-measure not appreciable. We truly appreciate you giving us the opportunity to rethink the phenomena behind the numerical values. We have included these analyses in the supplementary material in the hope that this will stimulate more thinking.

---

> > ### Author Response · Authors · 2023-08-23
> > **Response to Reviewer qgpw (Part 2/2)**
> >
> > ***Limitations:*** *This paper proposes an RGB-D video salient object detection dataset, which fills a research gap in the field and promotes the development of related studies. It has no potential negative societal impact. Furthermore, the authors have effectively addressed the limitations of their work. One suggestion here is to explore further to address the inconsistency between salient object patterns when they differ in depth images and RGB images. For example, an object may become salient in the depth image due to its close proximity, while in the RGB image, another object with brighter colors might be considered salient. Further exploration can provide insights into resolving this inconsistency between the two salient object modes.*
> >
> > **Answer:** Thanks for your valuable suggestion. The issue you highlighted is also a critical area of research in the field of RGB-D SOD to investigate how to effectively fuse inconsistent features across different modalities. In our DVSOD baseline, we address this challenge through the introduction of the Cross-Reference Module (CRM) [23], which employs an attention scheme to selectively aggregate crucial information from both RGB and depth streams. By properly weighting these features, our DVSOD baseline is able to produce a relatively reliable saliency map that works coherently across different imaging modes. We also come up with a promising future direction to further reconcile inconsistencies between two imaging modalities. We may develop a generative model in which features extracted from one modality could be used to generate a matched pseudo image of another modality, which is expected to harmonize the feature sets extracted from both modalities, thereby minimizing their divergences. We intend to delve into this direction in our forthcoming research.
> >
> > ------
> >
> > **We greatly appreciate the chance you've provided to enhance our paper through your insightful feedback and questions. We have carefully improved the paper accordingly (marked blue).**

---

### Official Review · Reviewer_4mrx · 2023-07-21
**Initial review**

**Rating:** 7
**Confidence:** 4
**Clarity:** Yes. The paper is well written and ea…

**Strengths:**

The dataset collected is large-scale, diverse, and of high-quality. The paper states clearly that the annotations are made using LabelMe, and other manual corrections to ensure correctness. The proposed baseline DVSOD makes sense to fuse RGB, depth and temporal information. The benchmark experiment is inclusive and demonstrates the usefulness of including more modality.

**Additional Feedback:**

I have no additional feedback.

**Correctness:**

The dataset is constructed in a sound way. The benchmark experiment is designed appropriately and analyzed well.

**Documentation:**

The provided URL works and I am able to access the dataset as well as baseline repository on github.

**Limitations:**

If possible, the dataset may add category label to the annotated instances. Also, if 3D annotation could be annotated or generated automatically, the dataset might benefit downstream tasks like 3D detection, object pose estimation, tracking, or reconstruction.

It seems the effort to collect such a dataset is huge (~13 months stated in supplementary). Could we utilize recent advances of photorealistic rendering to simulate data, or other media types like animation, to create annotations quicker, and evaluate sim2real using the proposed dataset?



**Opportunities For Improvement:**

I don't have major concerns about the paper. I will share some points of future extensions in limitation section.



**Relation To Prior Work:**

Yes. The paper is the first large-scale RGB+depth+video dataset for SOD. The contribution is clear.

**Summary And Contributions:**

The paper contributes a large-scale dataset for salient object detection (SOD) from RGB-D videos. The dataset fills the gap of no existing RGB+depth+video dataset on SOD. The dataset is diverse and inclusive, and of high quality with careful manual corrections during annotation. The benchmarking experiments involves enough baselines, and demonstrated the performance improvement by including depth and video temporal information. Potential research directions are listed and analyzed well in the end.

---

> ### Author Response · Authors · 2023-08-23
> **Response to Reviewer 4mrx**
>
> **Dear Reviewer 4mrx, we sincerely thank you for thoroughly reviewing our paper and providing these encouraging comments. Kindly find our detailed response to your valuable comments below.**
>
> ------
>
> ***Question1:*** *If possible, the dataset may add category label to the annotated instances. Also, if 3D annotation could be annotated or generated automatically, the dataset might benefit downstream tasks like 3D detection, object pose estimation, tracking, or reconstruction.*
>
>
> **Answer1:** We sincerely value your insightful feedback. Following your suggestions, we have augmented the proposed DViSal dataset by incorporating category labels for each salient instance, and have released them in our dataset's GitHub repository. The dataset now features 76 unique salient categories, and we hope this extension can open up new opportunities for further research and exploration.
>
> Furthermore, we have also taken the initiative to expand the data into 3D point clouds. From the sourced repositories, two datasets, Track3D [66] and Scene [28], have released their 3D point cloud data on their respective project websites. For other datasets, we provide conversion code that allows for the projection of RGB-D images into the 3D space. The code is accessible in our dataset's GitHub repository (link: https://github.com/DVSOD/DVSOD-DViSal -> Additional Resources). For each dataset we utilized, the necessary camera intrinsic parameters are generally available on their respective project websites. An exception is the DET dataset [64]; we attempted to contact the authors for the intrinsic camera parameters but learned that the authors have since left their academic institution, thus the camera's intrinsic parameters are currently unavailable. We anticipate that this endeavor will benefit downstream tasks, such as point cloud saliency detection, or tracking.
>
> ------
>
> ***Question2:*** *It seems the effort to collect such a dataset is huge (~13 months stated in supplementary). Could we utilize recent advances of photorealistic rendering to simulate data, or other media types like animation, to create annotations quicker, and evaluate sim2real using the proposed dataset?*
>
> **Answer2:** We appreciate your valuable recommendation. The ideas of utilizing simulate data to relieve labeling cost and improve the performance in real data are really interesting and worthy to explore.
>
> We attempt to study recent photorealistic rendering techniques and utilize them to simulate data automatically. To be specific, we establish an indoor synthetic RGB-D video SOD dataset by BlenderProc2 [ref1] using SceneNet [ref2]. This dataset includes 40 synthetic RGB-D videos accompanied with 2,060 salient object detection masks. We use point light sources as the type of light to simulate the lighting conditions. We also introduce randomly generated camera trajectories to achieve lifelike camera movement. Textures and materials are randomized.
>
> Based on this synthetic dataset, we carry out two exploratory experiments on sim2real, as outlined in the Table below. Model-A is trained on the synthetic dataset and tested on the proposed real-world dataset, and Model-B is pre-trained on the synthetic data, fine-tuned on our real-world training dataset, and evaluated on the real-world testing dataset. Our observations indicate that Model-A experiences a significant performance drop and Model-B keeps the original level or achieves marginal improvements. These outcomes might be due to two primary factors: 1) insufficient diversity in the synthetic dataset, and 2) the limited volume of simulated data, as rendering each video is time-intensive. In the future, we will continue to expand this synthetic dataset and we encourage researcher to investigate this interesting sim2real problem for DVSOD.
>
> [ref1] Blenderproc2: A procedural pipeline for photorealistic rendering. Journal of Open Source Software, 2023.
> [ref2] Understanding real world indoor scenes with synthetic data. CVPR, 2016.
>
> |   |  $E_{\xi}$     |  $S_{\alpha}$     |  $F_{\beta}$     |  $\mathcal{M}$     |
> |----------|-------|-------|-------|-------|
> | Baseline | 0.807  | 0.729  | 0.610  | 0.113  |
> | Model-A  | 0.668 | 0.458 | 0.234 | 0.180 |
> | Model-B  | 0.810 | 0.735 | 0.610 | 0.111 |
>
>
> ------
>
> **Thanks again for your encouragement and insightful suggestions.**

---

> > ### Comment · Reviewer_4mrx · 2023-08-24
> >
> > I would thank the authors for their additional efforts to address my comments. My score will hold for 'accept' for the submission.

---

> > > ### Author Response · Authors · 2023-08-24
> > >
> > > We would like to express our gratitude to be recognized by the reviewer. Thanks again for your efforts and insightful suggestions towards improving our work.

---

### Official Review · Reviewer_Vq9N · 2023-07-22
**Comprehensible study on RGB-D video salient object detection, but lacking a thorough comparison with state-of-the-art methods**

**Rating:** 7
**Confidence:** 3
**Clarity:** The paper is well-written.

**Strengths:**

1. The paper is easy to follow.

2. The paper introduces a comprehensive dataset consisting of diverse RGB-D videos with detailed annotations, enabling further research in RGB-D video salient object detection.

3. Benchmarking experiments: The authors conduct benchmarking experiments using various SOD models, demonstrating the effectiveness of multimodal video input for salient object detection.

4. The paper highlights how integrating depth maps as additional input dramatically enhances the accuracy in localizing salient objects in complex scenes.

**Additional Feedback:**

Please refer to the section of weaknesses.

**Correctness:**

This paper provides the detailed information of data collection/annotation protocol. It is constructed in a possibly sound way in my opinion.

**Documentation:**

There is sufficient detail on data collection and organization.

**Ethics:**

I don't have any ethical concerns with this submission.

**Limitations:**

I haven't identified any limitations and potential negative societal impact in this work.

**Opportunities For Improvement:**

1. While the paper mentions benchmarking experiments using various SOD models, it does not explicitly mention how these results compare to state-of-the-art methods in RGB-D video salient object detection (e.g., [1,2,3]). It would be helpful if the authors provided a detailed comparison with existing techniques. Also, it would be better to incorporate these methods to the related work section.

2. The annotation process involves multiple annotators selecting candidate salient objects based on their initial instinct, followed by a majority voting strategy to finalize the salient objects. This subjective nature of annotation can introduce biases or inconsistencies in labeling.

3. The experiment section (quantitative results in Section 4.3) does not provide detailed explanations for why certain models perform better than others or discuss potential reasons behind their findings.

4. While discussing efficiency improvements through lightweight strategies like depth-wise separable convolution or neural architecture search techniques, there is no analysis provided regarding their impact on computational requirements such as memory usage or inference time.

References:
[1] Lee, M., Park, C., Cho, S., & Lee, S. (2022, October). Spsn: Superpixel prototype sampling network for rgb-d salient object detection. In European Conference on Computer Vision (pp. 630-647). Cham: Springer Nature Switzerland.
[2] Zhang, W., Jiang, Y., Fu, K., & Zhao, Q. (2021, July). BTS-Net: Bi-directional transfer-and-selection network for RGB-D salient object detection. In 2021 IEEE International Conference on Multimedia and Expo (ICME) (pp. 1-6). IEEE.
[3] Liu, N., Zhang, N., Wan, K., Shao, L., & Han, J. (2021). Visual saliency transformer. In Proceedings of the IEEE/CVF international conference on computer vision (pp. 4722-4732).

**Relation To Prior Work:**

Probably no. Please refer to the first point in the section of weaknesses.

**Summary And Contributions:**

The paper introduces an easy-to-follow study on RGB-D video salient object detection. A major strength lies in the creation of a comprehensive dataset comprising diverse RGB-D videos with detailed annotations, which facilitates further research in this domain. Benchmarking experiments with various salient object detection models demonstrate the effectiveness of multimodal video input, particularly when integrating depth maps, leading to improved localization accuracy in complex scenes. However, the paper lacks a detailed comparison with state-of-the-art methods in RGB-D video salient object detection and could benefit from incorporating these methods into the related work section. The subjective nature of the annotation process may introduce biases or inconsistencies, and the experiment section does not thoroughly explain performance differences among models. Furthermore, while discussing efficiency improvements, there is no analysis of their impact on computational requirements.

---

> ### Author Response · Authors · 2023-08-23
> **Response to Reviewer Vq9N**
>
> **Dear Reviewer Vq9N, we greatly appreciate the time and effort you have dedicated to providing insightful feedback on ways to strengthen our paper. It is with great pleasure that we make a point-by-point response to all the comments.**
>
> ------
>
> ***Question1:*** *While the paper mentions benchmarking experiments using various SOD models, it does not explicitly mention how these results compare to state-of-the-art methods in RGB-D video salient object detection (e.g., [1,2,3]). It would be helpful if the authors provided a detailed comparison with existing techniques. Also, it would be better to incorporate these methods to the related work section.*
>
> **Answer1:** Thank you for the valuable suggestion to improve the completeness of our benchmarking experiments. We have included the experimental results of these saliency models [1, 2, 3] in Table 3 of the supplementary material and have discussed them in the related work section. We kindly raise the reviewer's attention that these models are primarily designed for static RGB-D images, not applicable to the new DVSOD task.
>
> ------
>
> ***Question2:*** *The annotation process involves multiple annotators selecting candidate salient objects based on their initial instinct, followed by a majority voting strategy to finalize the salient objects. This subjective nature of annotation can introduce biases or inconsistencies in labeling.*
>
> **Answer2:** Thank you for your valuable feedback. We'd like to clarify that the salient object detection task, by its nature, deals with what attracts human attention, and therefore, a certain degree of subjectivity is inherent. To bolster the reliability of the proposed dataset, we followed the prominent SOD datasets [14, 50, 58] for annotation. Specifically, we included multiple annotators (five in the paper) and adopted majority voting strategy to reduce the influence of individual biases, where objects were only marked as `salient' when at least 4 out of the 5 annotators concur. In this way, outliers or ambiguous cases will be filtered out, thereby mitigating the subjective bias of a particular annotator. We hope this could address your question.
>
> ------
>
> ***Question3:*** *The experiment section (quantitative results in Section 4.3) does not provide detailed explanations for why certain models perform better than others or discuss potential reasons behind their findings.*
>
> **Answer3:** Thanks for your valuable comment. In Section 4.3 (quantitative results), our primary focus was to verify the benefits of incorporating RGB-D and temporal features to enhance per-frame segmentation accuracy. To this end, we designed three comparative experiments, using a consistent backbone to ensure fairness in comparisons. We started with an RGB base model CPD [62], and then sequentially integrated depth elements and temporal video data to measure the performance enhancements achieved with each added dimension. Considering that DVSOD is relatively new and its development is still at an early stage, we additionally present the segmentation results of related SOD/RGB-D SOD/VSOD models in Table 2, to provide observational reference for our readers.
>
> ------
>
> ***Question4:*** *While discussing efficiency improvements through lightweight strategies like depth-wise separable convolution or neural architecture search techniques, there is no analysis provided regarding their impact on computational requirements such as memory usage or inference time.*
>
> **Answer4:** Thanks for your comments. Following your suggestion, we conducted an exploratory experiment to evaluate the impact of lightweight strategy by replacing the conventional convolution in our network encoder with the more efficient depth-wise separable convolution. Our experiments showed a reduction in network parameters from 97.3M to 58.1M and memory usage from 17.15G to 16.21G, along with an increase in inference speed from 16.7 FPS to 21.5 FPS. However, this change did lead to a drop in performance, with the F-measure falling from 0.610 to 0.575. These findings underscore the potential of efficiency-enhancing strategies, but they also highlight the need to balance accuracy with efficiency in saliency model design. We've incorporated these findings in our updated paper, and hope this will inspire more future works.
>
> ------
>
> **Thank you again for giving us the opportunity to strengthen our paper with your valuable comments and queries. We have carefully improved the paper accordingly (marked blue).**

---

### Decision · Program_Chairs · 2023-09-22

**Decision:**

Accept (Poster)

**Comment:**

This paper introduces a new dataset for salient object detection (SOD) in RGB-D videos. The dataset is comprehensive, comprising 237 RGB-D videos from multiple sources, annotated with instance-level, bounding box level and weak (scribble) annotations. Furthermore, the paper establishes a baseline benchmark on this new dataset by evaluating several recent RGB, RGB-D, and video SOD methods, and analyzes the benefits of each modality.

Pros: The introduced video SOD dataset is large-scale, containing diverse RGB-D videos and comprehensive annotations. Existing research has focused on RGB, RGB-D, or RGB-temporal modalities separately for SOD. The availability of this dataset will address a gap in the field for fusion methods. Moreover, the paper presents a comprehensive baseline by benchmarking several existing methods and a simple video RGB-D fusion method to help understand the importance of different modalities.

During the rebuttal, the authors addressed reviewers' concerns regarding SOTA baselines, annotation ambiguity/process, and detailed the explanation of the experiments. Furthermore, they have added semantic/3D annotations to the dataset. Reviewers anonymously support accepting the paper, and the AC agrees.